# DVI: A Derivative-based Vision Network for INR

**Runzhao Yang**[1] **Xiaolong Wu**[2] **Zhihong Zhang**[1 3] **Fabian Zhang**[4] **Tingxiong Xiao**[1] **Zongren Li**[5] **Kunlun He**[5] **Jinli Suo**[1 6 7]

## Abstract

Recent advancements in computer vision have seen Implicit Neural Representations (INR) becoming a dominant representation form for data due to their compactness and expressive power. To solve various vision tasks with INR data, vision networks can either be purely INR-based, but are thereby limited by simplistic operations and performance constraints, or include raster-based methods, which then tend to lose crucial structural features and important information of the INR during the conversion process. To address these issues, we propose DVI, a novel Derivative-based Vision network for INR, capable of handling a variety of vision tasks across various data modalities, while achieving the best performance among the existing methods by incorporating state of the art raster-based methods into a INR based architecture. DVI excels by leveraging the valuable features captured in the high order derivative map of the INR, then seamlessly fusing them into a pre-existing raster-based vision network, enhancing its performance with additional, task-relevant structural information. Extensive experiments on five vision tasks across three data modalities demonstrate DVI's superiority over existing methods. Additionally, our study encompasses comprehensive ablation studies to affirm the efficacy of each element of DVI, the influence of different derivative computation techniques and the impact of derivative orders. Reproducible codes are provided in the supplementary materials.

## 1. Introduction

Implicit Neural Representation (INR) is a novel form of data representation that models data through a mapping from coordinates to values. Unlike traditional raster representations, INR has the capability to model complex structural patterns and relationships within the data (Xu et al., 2022; Costain et al., 2023; Zhou et al., 2023a; De Luigi et al., 2023; Ramirez et al., 2023; Navon et al., 2023; Zhou et al., 2023b), making it extensively applicable in various vision data representations such as images (Strümpler et al., 2022), 3D volumes (Wu et al., 2021), and videos (Sitzmann et al., 2020). This enhanced capacity to model complex structural features makes INR particularly suited for tasks where traditional pixel or voxel representations are limited by resolution and scalability.

For performing specific vision tasks on the data in INR form, vision networks are required. These can be seperated into two categories, depending on the necessity to convert the data into raster form or not. Raster-based methods utilize pre-existing vision networks, on the other hand purely INR-based approaches operate solely in the INR domain. Raster-based methods involve converting INR back to raster form, subsequently employing a pre-existing vision network to execute the vision tasks. This approach effectively leverages the vast repository of existing algorithms. In contrast, standalone INR-based methods extract the structural information directly from the INR for visual tasks without converting it back to the raster form, which can save memory and hard disk bandwidth.

Currently, both approaches have significant drawbacks. Raster-based methods tend to lose crucial structural information modeled in the INR during the conversion process, limiting their performance in vision tasks. On the other hand, INR-based methods also face challenges due to their relative novelty, resulting in fewer existing vision networks that can serve as references. This often confines INR approaches to simpler architectures, compared to the vast array of sophisticated methods developed for raster-based processing, potentially hindering their adaptability and effectiveness for a broader range of complex vision tasks. Consequently, this can lead to limitations such as applicability to primarily simpler vision tasks and the reliance on structure-specific INR models that may not generalize well. These issues will

---

[1]Department of Automation, Tsinghua University, Beijing, China [2]Institute of Advanced Technology, University of Science and Technology of China, Hefei, China [3]Xiaomi Corporation, Shanghai, China [4]Department of Computer Science, ETH, Zurich, Switzerland [5]The People's Liberation Army General Hospital, Beijing, China [6]Institute of Brain and Cognitive Sciences, Tsinghua University, Beijing, China [7]Shanghai Artificial Intelligence Laboratory, Shanghai, China. Correspondence to: Jinli Suo <jl-suo@tsinghua.edu.cn>.

*Proceedings of the 42nd International Conference on Machine Learning*, Vancouver, Canada. PMLR 267, 2025. Copyright 2025 by the author(s).

be explored in the following sections.

To address the issues of both approaches, we propose DVI, a Derivative-based Vision network for INR, capable of handling a variety of vision tasks across various data modalities, while achieving the best performance among existing methods. Specifically, DVI firstly transforms INR data into a raster form, harnessing the strengths of pre-existing vision networks. Simultaneously, DVI extracts the structural information from a high order derivative map of the INR. This information is then seamlessly fused into the vision network, enhancing its performance with additional, task-relevant structural features that improve task outcomes.

We evaluate our method on five different vision tasks across three data modalities, demonstrating its superiority over the existing methods through extensive experiments on various datasets. Additionally, our research includes comprehensive ablation studies to validate the effectiveness of each component of our proposed method and explores the impact of different derivative computation techniques and derivative orders in our approach.

## 2. Related Work

### 2.1. Raster Representation

For a vision data with shape $s_1 \times \cdots \times s_n$ and $c$ channels, we typically represent it as an $n + 1$ dimensional array $\mathbf{D} \in \mathbb{R}^{s_1 \times \cdots \times s_n \times c}$ in raster form. With this representation, we can obtain a coordinate map $\mathbb{X} := \{\mathbf{x} | x_1 \in \{1..s_1\}, \ldots, x_n \in \{1..s_n\}\}$, corresponding to the data shape. The data values at any coordinate $\mathbf{x}$ can be fetched by indexing directly from the array as $\mathbf{D}[x_1, \ldots, x_n]$.

### 2.2. Implicit Neural Representation

In the realm of implicit neural representation (INR), we approach data representation through a fundamentally different lens. Unlike raster form, INR employs a neural network, denoted as the function $\mathcal{F} : \mathbb{X} \to \mathbb{R}^c$, mapping coordinates to data values, offering a more dynamic and potentially richer data interpretation. For optimal representation accuracy, the best $\mathcal{F}$ can be found by solving the following optimization problem:

$$\min_{\mathcal{F}} \quad \sum_{\mathbf{x} \in \mathbb{X}} \mathcal{L}(\mathcal{F}(\mathbf{x}), \mathbf{D}[\mathbf{x}]), \tag{1}$$

where $\mathcal{L}(\cdot)$ measures the representation accuracy. With this representation, we can fetch data values at any coordinate $\mathbf{x}$ by inputting it into $\mathcal{F}$ as $\mathcal{F}(\mathbf{x})$. So for any INR $\mathcal{F}$, we can easily convert it back to raster structure as $\widehat{\mathbf{D}} = \mathcal{F}(\mathbb{X})$. Currently, INRs have been extensively applied in a multitude of modalities, such as for the representation of images (Strümpler et al., 2022; Shen et al., 2022; Dupont et al.,

2021a; Sitzmann et al., 2020; Chen et al., 2021b; Saragadam et al., 2022), 3D volumes (Wu et al., 2021; Saragadam et al., 2022; Peng et al., 2020; Yariv et al., 2021; Takikawa et al., 2021), and video data (Sitzmann et al., 2020; Chen et al., 2021a; Saragadam et al., 2022; Chen et al., 2022; Mai & Liu, 2022). In those modalities, INRs perform various tasks, including rendering (Corona-Figueroa et al., 2022; Wang et al., 2022; Fang et al., 2022; Saragadam et al., 2022; Sitzmann et al., 2020; Takikawa et al., 2021; Qiu et al., 2023), registration (Li et al., 2024b; Wolterink et al., 2022; Byra et al., 2023; Zimmer et al., 2023; Sideri-Lampretsa et al., 2024; van Harten et al., 2024), and compression (Yang et al., 2023; Yang, 2023; Yang et al., 2024; Li et al., 2024a; Dupont et al., 2021a; Guo et al., 2024; Pistilli et al., 2022; Zhang et al., 2021c; Lee et al., 2021; Kwan et al., 2024).

### 2.3. Vision Tasks

Vision tasks can be divided into pixel-wise and image-wise categories. Pixel-wise vision tasks refer to tasks with finer granularity, where each pixel or voxel corresponds to a specific outcome, such as super-resolution (Image SR) (Lim et al., 2017; Liang et al., 2021), denoising (Image DN) (Zhang et al., 2017), segmentation (Volume Seg.) (Milletari et al., 2016; Çiçek et al., 2016), deblurring (Video DB) (Cao et al., 2023; Son et al., 2021), and optical flow estimation (Video FE) (Huang et al., 2022; Zhang et al., 2021a). For these tasks, numerous neural networks have been developed that process data in raster form. Our proposed method targets these pixel-wise vision tasks, aiming to extract structural information from the INR to enhance the performance of pre-existing raster-based networks. However, current INR-based vision networks are limited to basic operations such as interpolation and filtering (Xu et al., 2022; Nsampi et al., 2023), which often results in suboptimal performance in these detailed, pixel-wise vision tasks.

### 2.4. INR-based Vision Network

Many standalone INR methods which are not utilizing raster based methods have been proposed, however they face numerous challenges. (Cardace et al., 2024) proposed a novel network capable of directly processing structure-specific INR for segmentation or classification. Several works (Schürholt et al., 2021; Dupont et al., 2021b; 2022; Berardi et al., 2022; Schürholt et al., 2022; You et al., 2023; Lee et al., 2023; Bauer et al., 2023) introduced a generative model for INR by modeling the latent space of INR parameters, which is capable of handling basic completion and classification tasks. Further works (Zhou et al., 2023a; De Luigi et al., 2023; Ramirez et al., 2023; Navon et al., 2023; Zhou et al., 2023b) designed an encoder that converts all parameters in an INR into a feature vector for subsequent vision tasks. The trained encoder is only suitable for INR with a fixed number of parameters, yet vision data often

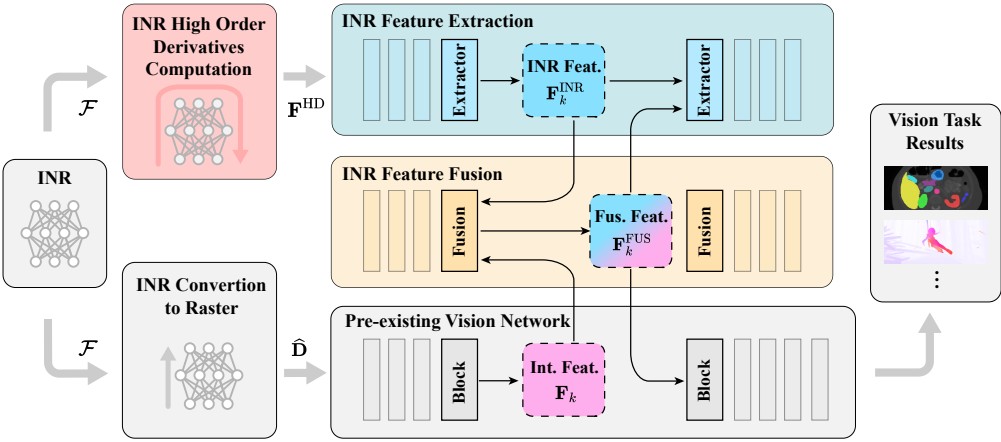

*Figure 1.* The overall architecture of the proposed DVI. Abbreviations stand for: Feat.: *Feature*, Fus.: *Fused*, Int.: *Intermediate*.

necessitates INR with varying parameters to accommodate different resolutions or demands. Following methods (Xu et al., 2022; Nsampi et al., 2023) extract structural information from the high order derivative of INR, capable of handling various vision tasks without restrictions on INR structure. However, without the use of conventional raster based networks, it is difficult to achieve good performance. Our approach DVI draws on these methods for extracting structural information from high order derivatives, and proposes a progressive fusion strategy to fuse the structural information with pre-existing raster-based vision networks to achieve the best performance.

## 3. Methodology

In this section, we delve into the methodologies for extracting structural information from Implicit Neural Representations (INR) and fusing this information into pre-existing raster-based vision networks. We begin by introducing the architecture and pipeline of our model DVI, outlining its innovative feature extraction and fusion strategy. Finally, we discuss in detail the critical designs of DVI.

### 3.1. Overall Architecture

The overall architecture of our method DVI, as depicted in Figure 1, comprises four primary components: 1) Pre-existing Vision Network, 2) INR High Order Derivative Computation, 3) INR Feature Extraction, and 4) INR Feature Fusion. As the first step, we convert INR to raster form, enabling the utilization of pre-existing vision networks. This ensures compatibility with established methods and harnesses their proven capabilities. Concurrently, we extract structural information from the high order derivative map of the INR. This information is then integrated into the vision network using the INR feature fusion module.

This strategy not only maintains the richness of the original INR data, but also augments the existing network's performance by infusing it with additional, task-relevant structural features that have been shown to improve task outcomes.

### 3.2. Pipeline

Please find symbols definition in the begining of the background section. Initially, we select a pre-existing vision network $\mathcal{H}(\cdot)$ for a given vision task that takes a raster representation as input and the task result as output. There are no restrictions on the network architectures, and we verified this by choosing several different networks in our experiments. At the beginning of the task, we will first transform the data from the INR $\mathcal{F}$ into raster structure:

$$\widehat{\mathbf{D}} = \mathcal{F}(\mathbb{X}), \qquad (2)$$

which will be fed into the vision network later. Simultaneously, DVI computes the high order derivative map of INR, encapsulating the structural information in

$$\mathbf{F}^{\text{HD}} = \mathcal{G}^{\text{HD}}(\mathcal{F}), \qquad (3)$$

where $\mathbf{F}^{\text{HD}} \in \mathbb{R}^{c_{HD} \times s_1 \times \cdots \times s_n}$ represents $c_{HD}$ partial derivatives of $\mathcal{F}$ at each point in $\mathbb{X}$, and $\mathcal{G}^{\text{HD}}(\cdot)$ is a specialized module that efficiently computes these derivatives, overcoming the limitations of traditional autograd methods (used in (Xu et al., 2022)) in terms of speed for higher order computations.

Next, DVI implements a progressive INR feature extraction and fusion strategy, designed to extract and integrate multiple levels of features from the INR into the vision network. This process involves using a set of $K$ INR feature extractors, $\{\mathcal{G}_k^{\text{INR}}(\cdot)\}_{k=1}^{K}$, to sequentially derive $K$ distinct features $\{\mathbf{F}_k^{\text{INR}}\}_{k=1}^{K}$ from the derivative map $\mathbf{F}^{\text{HD}}$. These

features are represented as:

$$\mathbf{F}_1^{\text{INR}} = \mathcal{G}_1^{\text{INR}}(\mathbf{F}^{\text{HD}}), \tag{4}$$

$$\mathbf{F}_{k+1}^{\text{INR}} = \mathcal{G}_{k+1}^{\text{INR}}(\mathbf{F}_k^{\text{INR}}, \mathbf{F}_k^{\text{FUS}}) \quad \text{for } k = 1, \dots, K-1, \tag{5}$$

where $\mathbf{F}_k^{\text{FUS}}$ is a feature fused from the previous level. Using the fused features from the previous level as additional inputs to the next feature extractor helps to extract features that are aligned with the target vision task. We aimed to align the INR's structural features with the target network's features in multi-level, facilitating the extraction of structural information with varying densities relevant to the visual task.

Further, the outputs from $K$ distinct layers of the vision network $\mathcal{H}(\cdot)$ are selected as intermediate features, denoted as $\{\mathbf{F}_k\}_{k=1}^K$. For simplicity, we segment $\mathcal{H}$ into $K+1$ sequential blocks based on the positions of these $K$ features, denoted as $\mathcal{H} = \mathcal{H}_1 \circ \cdots \mathcal{H}_K \circ \mathcal{H}_{K+1}$.

Finally, the extracted INR features $\{\mathbf{F}_k^{\text{INR}}\}_{k=1}^K$ are fused with the corresponding intermediate features $\{\mathbf{F}_k\}_{k=1}^K$ of the vision network into the fused features $\mathbf{F}_k^{\text{FUS}}$. This is achieved through a series of $K$ INR feature fusion modules, $\{\mathcal{G}_k^{\text{FUS}}(\cdot)\}_{k=1}^K$, which are employed successively:

$$\mathbf{F}_k^{\text{FUS}} = \mathcal{G}_k^{\text{FUS}}(\mathbf{F}_k^{\text{INR}}, \mathbf{F}_k) \quad \text{for } k = 1, \dots, K. \tag{6}$$

The fused features $\mathbf{F}_k^{\text{FUS}}$ have the same shape as the intermediate features $\mathbf{F}_k$, and will replace them as the input to the subsequent vision network block. This progressive fusion not only aligns, but also enriches the network's intermediate features with the structural feature captured from the INR.

The overall pipeline becomes:

$$\mathbf{F}_1 = \mathcal{H}_1(\widehat{\mathbf{D}}), \tag{7}$$

$$\mathbf{F}_{k+1} = \mathcal{H}_{k+1}(\mathbf{F}_k^{\text{FUS}}) \quad \text{for } k = 1, \dots, K-1, \tag{8}$$

$$\text{Task Results} = \mathcal{H}_{K+1}(\mathbf{F}_K^{\text{FUS}}). \tag{9}$$

The progressive strategy adopted by DVI offers remarkable flexibility, adapting seamlessly to a wide range of pre-existing vision networks. This approach not only facilitates the effective extraction of structural information from INR, but also ensures its precise integration into the network's processing flow. In the subsequent sections, we will delve into the details of DVI.

### 3.3. INR High Order Derivative Computation

Let the vector consisting of all $pth$ order derivatives of $\mathcal{F}$ with respect to a point $\mathbf{x} \in \mathbb{X}$ be denoted by:

$$\mathfrak{D}_{\mathbf{x}}^p \mathcal{F}(\mathbf{x}) :=$$
$$\left[ \frac{\partial^p f(\mathbf{x})_1}{\partial x_1^p}, \frac{\partial^p f(\mathbf{x})_1}{\partial x_1^{p-1} \partial x_2}, \cdots, \frac{\partial^p f(\mathbf{x})_c}{\partial x_{n-1} \partial x_n^{p-1}}, \frac{\partial^p f(\mathbf{x})_c}{\partial x_n^p} \right]^\mathsf{T}, \tag{10}$$

which contains $c * \binom{n+p-1}{p}$ partial derivatives. Then, the derivative map we need to compute can be represented as a tensor consisting of the $1^{st}$ to $P^{th}$ order derivatives at each point:

$$\mathbf{F}^{drv} = \left( \begin{bmatrix} \mathfrak{D}_{\mathbf{x}}^1 \mathcal{F}(x_1, \cdots, x_n) \\ \vdots \\ \mathfrak{D}_{\mathbf{x}}^P \mathcal{F}(x_1, \cdots, x_n) \end{bmatrix} \right)_{s_1 \times \cdots \times s_n}. \tag{11}$$

Computing all these derivatives using autograd would be highly time consuming. We use the recursive formula for high order derivatives in (Xiao et al., 2023) to compute the derivative map at an accelerated speed.

Due to the large difference in values between different order derivatives, we need to normalize them. We first counted the distribution of each order derivatives on the training set, (which was found to be approximated as a 0-mean Gaussian distribution), computed the maximum value, and normalized each order derivatives by their corresponding maximum value.

### 3.4. INR Feature Extraction

The INR feature extraction in DVI employs a series of $K$ extractors, all sharing a similar structure, with the exception of the first extractor, which lacks a residual connection at its entrance. As illustrated in Figure 2(a) and (b), each INR feature extractor is comprised of residual connections, Swin Transformer layers (STL), and convolutional layers (CONV). The specific calculation process for the feature extraction is:

$$\begin{aligned} \mathbf{F}_{k+1}^{\text{INR}} &= \mathcal{G}_{k+1}^{\text{INR}}(\mathbf{F}_k^{\text{INR}}, \mathbf{F}_k^{\text{FUS}}) \\ &= \mathcal{G}^{\text{CONV}} \circ \mathcal{G}^{\text{STL}} \circ \cdots \circ \mathcal{G}^{\text{STL}} \circ \\ &\quad \mathcal{G}^{\text{CONV}}(\mathbf{F}_k^{\text{INR}} + \mathbf{F}_k^{\text{FUS}}) + \mathbf{F}_k^{\text{INR}}. \end{aligned} \tag{12}$$

The feature extraction process involves residual connections at both the entrance and exit of the extractor. The entrance residual connection robustly incorporates the fused feature from the previous level, enhancing the stability of feature introduction, as seen in (He et al., 2016). The exit residual connection on the other hand, aggregates features from each level, contributing to a more coherent feature extraction

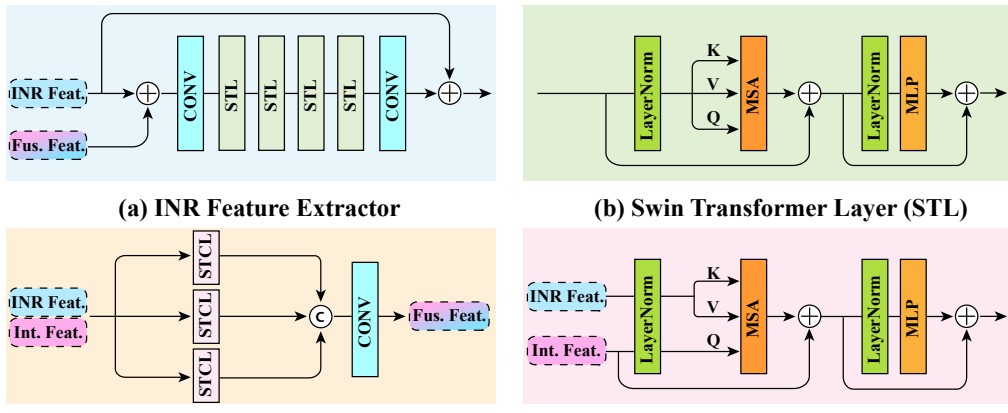

**(a) INR Feature Extractor**   **(b) Swin Transformer Layer (STL)**

**(c) INR Feature Fusion Module**   **(d) Swin Transformer Cross-attention Layer (STCL)**

*Figure 2.* The architectures of (a) INR feature Extractor, (b) Swin Transformer Layer, (c) INR Feature Fusion Module, and (d) Swin Transformer Cross-attention Layer. Abbreviations stand for: Feat.: *Feature*, Fus.: *Fused*, Int.: *Intermediate*.

(Liang et al., 2021). CONVs are positioned at the entrance and exit as well. The entrance CONV facilitates early visual processing (Xiao et al., 2021), while the exit CONV enhances the extractor's translational equivariance (Liang et al., 2021).

At the heart of the extractor lies the STL (Liu et al., 2021; Liang et al., 2021), which shares structural similarities with traditional Transformer architectures, comprising of Layer Normalization (LN), Multi-Head Self-Attention (MSA), and a Multi-Layer Perceptron (MLP). A distinctive feature of STL is its approach to attention computation within the MSA. STL partitions input features into non-overlapping windows and computes standard self-attention separately within these windows as:

$$\mathbf{Q} = \mathbf{F}^{\text{WIN}}\mathbf{W_Q}, \mathbf{K} = \mathbf{F}^{\text{WIN}}\mathbf{W_K}, \mathbf{V} = \mathbf{F}^{\text{WIN}}\mathbf{W_V}, \quad (13)$$

$$\text{Attention}(\mathbf{Q}, \mathbf{K}, \mathbf{V}) = \text{SoftMax}(\mathbf{Q}\mathbf{K}^{\mathsf{T}}/\sqrt{d} + \mathbf{B})\mathbf{V}, \quad (14)$$

where $\mathbf{F}^{\text{WIN}}$ denotes the features within each window, $\mathbf{W_Q}$, $\mathbf{W_K}$, and $\mathbf{W_V}$ are the shared projection matrices, $d$ is the feature dimension, and $\mathbf{B}$ represents a learnable relative positional encoding. To facilitate cross-window attention, the STL alternates between regular and shifted window partitioning in its MSA, as proposed in (Liu et al., 2021).

### 3.5. INR Feature Fusion

The INR feature fusion in DVI consists of $K$ identical fusion modules. Each module is characterized by three Swin Transformer Cross-attention layers (STCL) and a convolutional layer (CONV), as depicted in Figure 2(c) and (d). Each STCL closely mirrors the structure of the STL, with a notable distinction being the incorporation of cross-attention within the MSA. This adaptation is pivotal for the feature fusion process.

The fusion of $\mathbf{F}_k^{\text{INR}}$ into $\mathbf{F}_k$ is accomplished by mapping $\mathbf{F}_k$ to the query and $\mathbf{F}_k^{\text{INR}}$ to both key and value in the cross-attention framework. The cross-attention computation follows the formula outlined in Equation (14).

A distinguishing feature of all three STCLs is their use of regular window partitioning within the MSA, albeit with varying window sizes. This allows DVI to execute feature fusion across three distinct spatial scales. Once the fused features are computed at these varying scales, they are concatenated along the channel dimension. Subsequently, their dimensionality is adjusted to align with that of the intermediate feature using a $1 \times 1$ CONV. The specific calculation process for the feature fusion is:

$$\begin{aligned} \mathbf{F}_k^{\text{FUS}} &= \mathcal{G}_k^{\text{FUS}}(\mathbf{F}_k^{\text{INR}}, \mathbf{F}_k) \\ &= \mathcal{G}^{\text{CONV}} \circ \text{C}(\mathcal{G}_{ws=2}^{\text{STCL}}(\mathbf{F}_k^{\text{INR}}, \mathbf{F}_k), \\ &\quad \mathcal{G}_{ws=4}^{\text{STCL}}(\mathbf{F}_k^{\text{INR}}, \mathbf{F}_k), \mathcal{G}_{ws=8}^{\text{STCL}}(\mathbf{F}_k^{\text{INR}}, \mathbf{F}_k)), \end{aligned} \quad (15)$$

where $\text{C}(\cdot)$ denotes the channel concatenation operator and $ws$ represents the size of the regular window partitioning.

## 4. Experiments

We evaluate our proposed DVI on multiple vision tasks across three different types of data modalities: 1) Image Tasks; 2) 3D Volume Tasks and 3) Video Tasks. The methods named 'DVI(*net*)' represents DVI with *net* as the respective pre-existing network. All details of data preparation and training from this section can be found in the supplementary materials.

## 4.1. Data

For the image super-resolution task, we adopted the setup from works (Lim et al., 2017; Liang et al., 2021; Li et al., 2023), utilizing DIV2K (Agustsson & Timofte, 2017) as the training set, with Set5 (Bevilacqua et al., 2012), Set14 (Zeyde et al., 2012), BSD100 (Martin et al., 2001b), Urban100 (Huang et al., 2015), and Manga109 (Matsui et al., 2017) serving as test sets. Similarly, for the image denoising task, the setup from works (Zhang et al., 2021b; Liang et al., 2021; Li et al., 2023) was followed, employing BSD500 (Martin et al., 2001a) and WED (Ma et al., 2016) as training sets, along with CBSD68 (Martin et al., 2001a), Kodak24 (Franzen, 1999), and McMaster (Zhang et al., 2011) as test sets. In the domain of 3D volume segmentation, the setup from works (Milletari et al., 2016; Çiçek et al., 2016), using Synapse (Landman et al., 2015), was followed. For video tasks, GoPro (Nah et al., 2017) was used as the benchmark for video deblurring, following the setup in works (Cao et al., 2023; Son et al., 2021), and Sintel (Butler et al., 2012) for video optical flow estimation, based on the methods described in (Huang et al., 2022; Zhang et al., 2021a).

All input data, including downsampled images for super-resolution, noise-added images for denoising, 3D volumes for segmentation, and videos for deblurring and optical flow estimation, were converted to Implicit Neural Representation (INR) form to standardize the data processing pipeline across different tasks.

## 4.2. Baselines

We distinguish between two types of approaches: Raster-based approaches, where we choose EDSR (Lim et al., 2017), SwinIR (Liang et al., 2021) and StableSR (Wang et al., 2024) as comparison algorithms for image super-resolution task, SwinIR (Liang et al., 2021) , DnCNN (Zhang et al., 2017) and DiffBIR (Lin et al., 2024) for image denoising task, VNET (Milletari et al., 2016), UNet3D (Çiçek et al., 2016) and MedSegDiff-V2 (Wu et al., 2023) for 3D volume segmentation task, VDTR (Cao et al., 2023), PVDNet (Son et al., 2021) and VD-Diff (Rao et al., 2024) for video deblurring task, FlowFormer (Huang et al., 2022), SepFlow (Zhang et al., 2021a) and FlowDiffuser (Luo et al., 2024) for video optical flow estimation task. For the INR-based approach, we use INSP (Xu et al., 2022) as the comparison for all vision tasks.

## 4.3. Main Results

Quantitative results are shown in Tables 1 to 4. Visual results are shown in Figures 3 and S1 to S5. Following observations can be made: 1) Our approach DVI consistently outperforms the raster-based approaches and the INR-based on all vision tasks. 2) The INSP method is not suitable for performing the complex vision tasks, except for 3D volume segmentation. It should be noted that this comparison is influenced by fundamental methodological differences: INSP tackles a more challenging problem by processing INRs solely through their weights without materializing discrete signals. 3) For the image super-resolution, 3D volume segmentation and video flow estimation tasks, the improvement of DVI is most pronounced compared to the raster-based methods, indicating that the structural information encoded in the INR is more helpful for these specific tasks. 4) DVI underperforms on tasks requiring coarse structural information, such as video classification. To improve performance, we can reduce the spatio-temporal resolution (res↓) to remove redundant information and increase the order of derivatives (rf↑) to expand the structural feature "receptive field." Testing on ViViT model with the Something-Something V2 dataset, as shown in Table 5, supports this approach.

*Table 1.* Quantitative results (PSNR↑ & SSIM↑) for image super-resolution task.

| Method | Set5 | | Set14 | | BSD100 | | Urban100 | | Manga109 | | MAC(G) | Param(M) |
|---|---|---|---|---|---|---|---|---|---|---|---|---|
| | PSNR↑ | SSIM↑ | PSNR↑ | SSIM↑ | PSNR↑ | SSIM↑ | PSNR↑ | SSIM↑ | PSNR↑ | SSIM↑ | | |
| INSP (Xu et al., 2022) | 19.37 | 0.6950 | 18.80 | 0.6202 | 20.07 | 0.6196 | 17.15 | 0.5348 | 14.63 | 0.5411 | 395 | 11 |
| EDSR (Lim et al., 2017) | 30.08 | 0.8509 | 27.24 | 0.7591 | 25.78 | 0.7614 | 23.49 | 0.7883 | 27.14 | 0.8643 | 1532 | 159 |
| **DVI(EDSR)** | **30.92** | **0.8769** | **28.09** | **0.7997** | **26.84** | **0.8051** | **24.71** | **0.8211** | **28.23** | **0.8856** | 1681 | 183 |
| SwinIR (Liang et al., 2021) | 30.00 | 0.8511 | 27.25 | 0.7604 | 25.66 | 0.7619 | 23.28 | 0.7815 | 27.02 | 0.8645 | 91 | 11 |
| **DVI(SwinIR)** | **31.96** | **0.9039** | **31.19** | **0.8548** | **27.53** | **0.8371** | **25.47** | **0.8513** | **29.18** | **0.9105** | 101 | 15 |
| StableSR (Wang et al., 2024) | 30.09 | 0.8516 | 27.25 | 0.7600 | 25.34 | 0.7602 | 23.18 | 0.7788 | 26.81 | 0.8550 | 12453 | 148 |
| **DVI(StableSR)** | **31.58** | **0.9001** | **31.12** | **0.8526** | **27.06** | **0.8195** | **25.12** | **0.8421** | **28.97** | **0.8973** | 12581 | 156 |

*Table 2.* Quantitative results (PSNR↑ & SSIM↑) for image denoising task.

| Method | Kodak24 | | CBSD68 | | McMaster | | MAC(G) | Param(M) |
|---|---|---|---|---|---|---|---|---|
| | PSNR↑ | SSIM↑ | PSNR↑ | SSIM↑ | PSNR↑ | SSIM↑ | | |
| INSP (Xu et al., 2022) | 23.46 | 0.7769 | 22.45 | 0.7848 | 22.43 | 0.7035 | 1034 | 4 |
| DnCNN (Zhang et al., 2017) | 29.13 | 0.7414 | 28.57 | 0.7635 | 28.73 | 0.7106 | 167 | 0.6 |
| **DVI(DnCNN)** | **31.97** | **0.8717** | **31.16** | **0.8770** | **31.25** | **0.8330** | 267 | 1 |
| SwinIR (Liang et al., 2021) | 34.53 | 0.9188 | 33.60 | 0.9242 | 34.87 | 0.9247 | 539 | 12 |
| **DVI(SwinIR)** | **35.95** | **0.9552** | **35.05** | **0.9465** | **36.26** | **0.9445** | 1071 | 15 |
| DiffBIR (Lin et al., 2024) | 34.34 | 0.9335 | 33.42 | 0.9202 | 33.98 | 0.9114 | 3596 | 379 |
| **DVI(DiffBIR)** | **35.53** | **0.9511** | **35.05** | **0.9480** | **35.79** | **0.9395** | 3690 | 385 |

*Table 3.* Quantitative results (DSC↑) for 3D volume segmentation task.

| Method | Mean | Spl | Rkid | Lkid | Gal | Liv | Sto | Aor | Pan | MAC(G) | Param(M) |
|---|---|---|---|---|---|---|---|---|---|---|---|
| INSP (Xu et al., 2022) | 45.97 | 38.53 | 28.85 | 35.58 | 53.5 | 50.87 | 73.12 | 45.73 | 41.6 | 45 | 0.2 |
| UNet3D (Çiçek et al., 2016) | 68.46 | 84.06 | 82.41 | 84.41 | 22.3 | 92.02 | 65.64 | 75.25 | 41.58 | 7 | 2 |
| **DVI(UNet3D)** | **80.49** | **85.17** | **89.31** | **87.54** | **51.06** | **92.75** | **79.38** | **92.52** | **66.19** | 15 | 2 |
| VNET (Milletari et al., 2016) | 72.62 | **86.27** | 86.42 | 85.64 | 34.71 | **93.16** | 70.39 | 74.99 | 49.38 | 31 | 11 |
| **DVI(VNET)** | **83.60** | 78.00 | **87.10** | **91.49** | **73.23** | 83.96 | **77.81** | **94.34** | **82.91** | 43 | 11 |
| MedSegDiff-V2 (Wu et al., 2023) | 75.79 | **86.35** | 85.31 | 87.25 | 48.39 | **89.55** | 73.40 | 75.36 | 60.67 | 1966 | 44 |
| **DVI(MedSegDiff-V2)** | **85.46** | 77.63 | **87.03** | **94.23** | **75.36** | 82.31 | **82.53** | **95.02** | **89.58** | 2101 | 46 |

*Table 4.* Left: Quantitative results (PSNR↑ & SSIM↑) for video deblurring task. Right: Quantitative results (EPE↓) for video optical flow estimation task.

| Method | GoPro | | | | Method | Sintel(final) | | | | |
|---|---|---|---|---|---|---|---|---|---|---|
| | PSNR↑ | SSIM↑ | MAC(G) | Param(M) | | all | matched | unmat. | MAC(G) | Param(M) |
| INSP (Xu et al., 2022) | 20.00 | 0.6449 | 400 | 2 | INSP (Xu et al., 2022) | 10.42 | 9.01 | 32.29 | 187 | 2 |
| PVDNet (Son et al., 2021) | 25.98 | 0.7993 | 250 | 10 | SepFlow (Zhang et al., 2021a) | 15.90 | 13.42 | 38.44 | 125 | 8 |
| **DVI(PVDNet)** | **27.09** | **0.8401** | 338 | 12 | **DVI(SepFlow)** | **9.35** | **7.90** | **22.48** | 178 | 16 |
| VDTR (Cao et al., 2023) | 26.79 | 0.7935 | 347 | 23 | FlowFormer (Huang et al., 2022) | 6.35 | 4.41 | 23.95 | 93 | 16 |
| **DVI(VDTR)** | **27.86** | **0.8458** | 367 | 30 | **DVI(FlowFormer)** | **5.67** | **3.86** | **22.00** | 139 | 24 |
| VD-Diff (Rao et al., 2024) | 28.23 | 0.8691 | 236 | 12 | FlowDiffuser (Luo et al., 2024) | 4.94 | 4.17 | 11.90 | 312 | 15 |
| **DVI(VD-Diff)** | **29.07** | **0.9006** | 259 | 13 | **DVI(FlowDiffuser)** | **3.92** | **3.31** | **9.45** | 341 | 16 |

Table 5. Quantitative results for video classification task.

| Method | ViViT | DVI(ViViT) | DVI(ViViT)+res↓ | DVI(ViViT)+res↓+rf↑ |
|---|---|---|---|---|
| Top1-accuracy↑ | 56.8 | **57.0** | **60.2** | **64.9** |

Figure 3. Visual comparisons. Please refer to Figures S1 to S5 for more results.

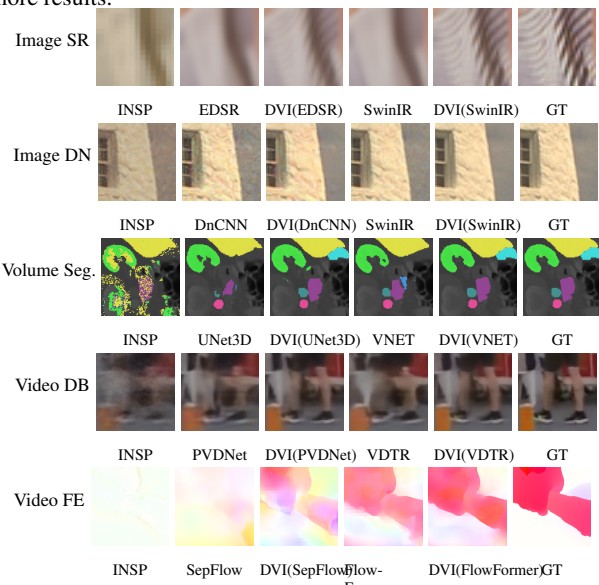

### 5. Analysis

We further verify the validity of various aspects of DVI and investigate the effect of different orders of derivatives on DVI. All implementation details are deferred to supplementary materials. Following distinct observations can be made:

#### 5.1. DVI is Robust to Various Pre-existing Network Architectures

Table 6 reveals a significant ($p < 0.05$) improvement in performance with our method compared to the respective pre-existing network, irrespective of the network architectures employed. This consistency underscores the robust nature of our method in diverse network architectures.

#### 5.2. Derivative Map Contains Task-Relevant Structural Features

Figure 4 shows that employing appropriate derivative maps substantially elevates performance over the pre-existing network. In contrast, a mismatched map can significantly reduce performance, and maps with zero or random values do not yield significant improvements. These findings suggest that derivative maps are integral to enhancing task performance, presumably due to their encapsulation of critical structural information. Please refer to Figure S6 for more details.

#### 5.3. Contribution of Feature Extraction and Fusion

Figure 5(a) shows the impact of removing the feature extraction and fusion modules from DVI. In the 'w/o E&F' setting, we removed the feature extraction and fusion modules and plainly fused the derivative map into the pre-existing network by concatenating them to the channel dimension of the input data, where the first layer of the pre-existing network was adjusted to fit the expanded channels. We find that even after removing the feature extraction and fusion modules, there is still some performance improvement, due to the structural information in the derivative maps. However, there is a significant decrease in performance compared to DVI. This fully demonstrates the importance of the feature extraction and fusion modules to DVI. Please refer to Figure S7 for more details.

#### 5.4. DVI is Better than INR-SR in Image Super-resolution

For the INR-SR approach, we achieve image super-resolution by supersampling the INR. Figure 5(b) demonstrates that our method DVI surpasses the INR-SR in terms of performance improvement relative to the pre-existing network. Please refer to Figure S7 for more details.

#### 5.5. Derivative Computation Techniques

Figure 6 shows that the performance of our method is comparable to autograd. However, as illustrated in the right panel, our method demonstrates a notable speed advantage over autograd, particularly when dealing with higher order derivatives. Please refer to Figure S8 for more details.

#### 5.6. Impact of the Highest Order of Derivative Map

Figure 7 shows the performance of DVI varies with the change in the highest order differently in the two tasks, which may suggest a different role for the derivative map in the two tasks. Also, in both tasks there was a significant drop in performance when the highest order reached 5, which may be due to the excessive redundancy of the derivative map affecting the training of the neural network. Please refer to Figure S9 for more details.

### 6. Discussions

#### 6.1. Computational Costs of DVI

During the training phase, our method requires additional computational overhead compared to pre-existing vision networks, primarily due to the need to train the INR feature extractors and fusion modules. In the inference stage, additional computational load mainly stems from the computation of derivative maps, the INR feature extraction and fusion network inference. For efficient computation

*Table 6.* Statistical significance of performance differences between DVI(*net*) and the respective pre-existing network *net* across different tasks.

| Task | Image SR | | Image DN | | Volume Seg. | | Video DB | | Video FE | |
|---|---|---|---|---|---|---|---|---|---|---|
| *net* | EDSR (Lim et al., 2017) | SwinIR (Liang et al., 2021) | DnCNN (Zhang et al., 2017) | SwinIR (Liang et al., 2021) | UNet3D (Çiçek et al., 2016) | VNET (Milletari et al., 2016) | PVDNet (Son et al., 2021) | VDTR (Cao et al., 2023) | SepFlow (Zhang et al., 2021a) | FlowFormer (Huang et al., 2022) |
| p-value | 3.3E-31 | 3.8E-04 | 5.6E-18 | 4.0E-04 | 1.0E-03 | 2.5E-03 | 2.8E-11 | 2.7E-08 | 2.4E-02 | 2.2E-02 |

*Figure 4.* Assessing the impact of substituting DVI's derivative map with alternative maps - zero-value (zero), random-value (random), and mismatched derivative (mismatch) - on the super-resolution task for the Manga109 dataset using SwinIR as the pre-existing network. On the left are bar plots for each alternative, with significant differences indicated (****: p < 0.0001). On the right are box plots showing the performance improvement of each alternative over the pre-existing network.

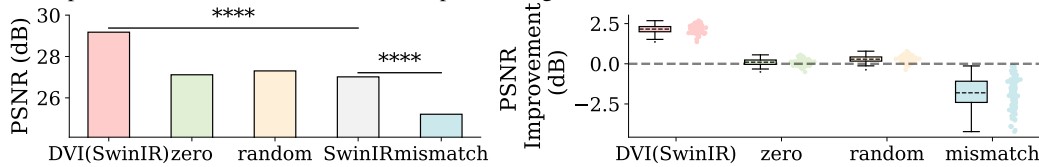

*Figure 5.* (a) The impact of removing the feature extraction and fusion modules from DVI on dennoising task, Kodak24 dataset, with DnCNN as pre-existing network. (b) The comparison between DVI and the super-resolution sampling technique using INR (INR-SR) on super-resolution task, Urban100 dataset, with SwinIR as pre-existing network. Both subfigures have the same layouts as Figure 4.

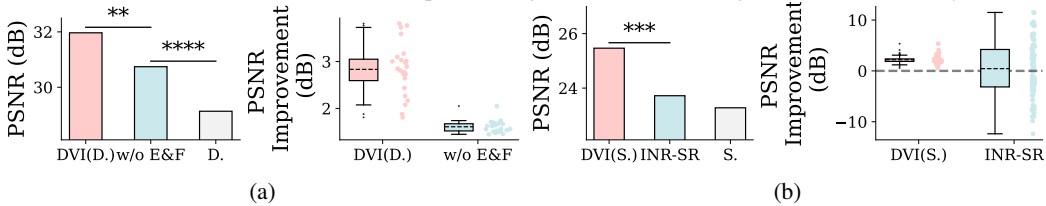

*Figure 6.* The performance comparison of DVI on 3D volume segmentation task employing two distinct derivative computation techniques with VNET as pre-existing network. Left: barplots of each techniques. Right: curves of time (network inference time + derivative map calculation time) vs. the highest order of the derivative map for each technique in log scale.

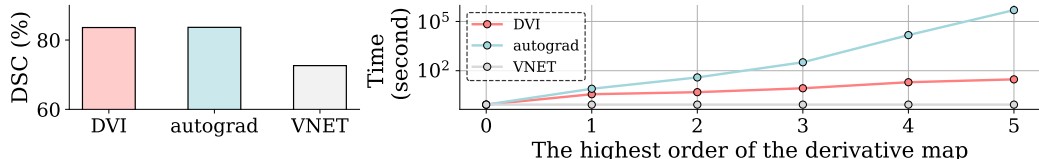

*Figure 7.* The curves of performance on video deblurring task (a) and video flow estimation task (b) versus the highest order of the derivative map employed in DVI.

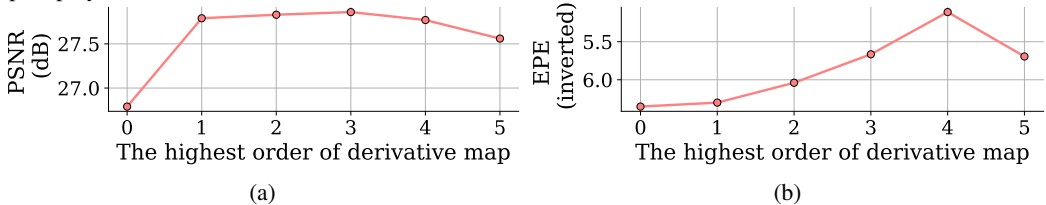

of derivative maps, our method has already achieved significant improvements over autograd. Further enhancements could potentially arise from more optimal choices of derivative map orders (discussed in detail in the following section) or through CUDA code restructuring. For efficient computation in the INR feature extraction and fusion network, future work could explore sparser feature fusion strategies (also discussed in the next section) or the adoption of more efficient neural network architectures.

## 6.2. Exploring More Rational Orders of Derivative Maps

The complexity of computing $1^{st}$ to $P^{th}$ order derivatives in our method is $\mathcal{O}(P^3) < \mathcal{T}_{ours}(P) << \mathcal{O}(n^P)$. Therefore, reducing the order $P$ can lower the complexity. We can identify the minimal order suitable for specific vision tasks through multiple experiments, thus ensuring efficient computation without compromising accuracy. Additionally, we can compute different orders of derivatives for different points. For example, we could estimate the error map of the pre-existing vision network (Selvaraju et al., 2017), and in areas with higher errors, compute higher order derivatives, while lower orders suffice in other regions. This approach could strike a better balance between performance and efficiency.

## 6.3. Exploring Sparser Feature Fusion Strategies

Our method separately fuses two feature maps within multiple non-overlapping windows. Reducing the number of windows can decrease complexity, as seen in (Liu et al., 2021). Thus, we could predict a highly sparse mask before feature fusion, then conducting feature fusion only within the windows covered by this mask. This strategy can potentially reduce computational demand while maintaining the integrity and effectiveness of the feature fusion process.

## 6.4. Data Augmentation

For simple data augmentation such as flipping, rotation, and cropping, we can obtain the augmented paired data (raster form and INR) on-the-fly by performing the same operation on the INR. However, for complex data augmentation such as color jittering, adding noise and scaling, we need to further investigate how to generate the corresponding INR on-the-fly. It is worth noting that although we removed these complex data augmentations in all experiments, we still achieved the best performance overall.

*Figure 8.* The performance comparison of DVI on segmentation task with NeRF-like methods.

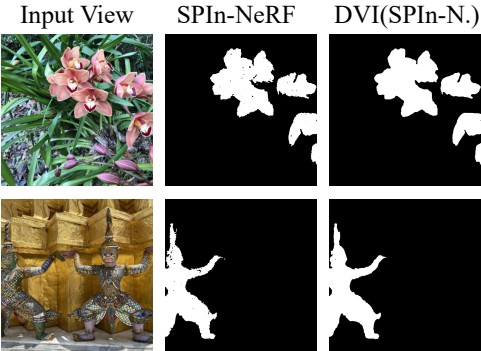

| Input View | SPIn-NeRF | DVI(SPIn-N.) |

*Table 7.* Quantitative results for 3D volume segmentation task with three different INRs and different derivatives calculation methods.

| Method | SIREN | | ReLU P.E. | | FFN | | Neumrical |
|---|---|---|---|---|---|---|---|
| | Ours | VNET | Ours | VNET | Ours | VNET | |
| DSC↑ | 83.60 | 72.62 | 80.29 | 71.03 | 80.00 | 68.52 | 74.33 |

## 6.5. Adapting DVI to non-CNN/transformer networks

DVI can enhance performance by extracting structural information from INRs, applicable to the algorithm using INR, including NeRF-like models such as SPIn-NeRF (Figure 8). We calculated first-order derivatives of the logit and density with respect to all feature embeddings and fed them into a new fully connected layer to predict the logit. As shown in Figure 8(b), DVI improves the accuracy and continuity of the segmentation mask.

## 6.6. The Performance of DVI on Other INRs

DVI achieves similar results on other INRs as long as higher-order derivatives can be computed, as shown in Table 7.

## 6.7. Using The Derivatives from The Raw Signal

Numerical derivatives from the raw signal are ineffective, as shown in the "Numerical" column of Table 7 compared to the "SIREN-Ours" column. The derivatives from INR are effective because they encode structural information during the fitting process.

# 7. Conclusions

Our study presents DVI, a Derivative-based Vision Network for INR, addressing the limitations of existing methods in handling vision tasks for INR. DVI excels by extracting structural information from INR's high order derivative map, enhancing the performance of an array of different pre-existing vision networks with deeper, task-specific insights. Extensive testing across various vision tasks and data modalities confirms DVI's superior performance over existing methods, proving the efficacy of our approach of fusing and harnessing strengths of both INR and raster-based methods.

## Acknowledgements

We would like to express our sincere gratitude to all reviewers, especially "Kpuk", for providing comprehensive and detailed suggestions that significantly improved this paper. We also extend our appreciation to Dr. Xia Li and Dr. Weijie Wang from ETH Zurich and Ph.D. candidate Qianni Cao from Tsinghua University for their valuable insights and constructive feedback throughout this research. This work is jointly supported by the National Key R&D Program of China (Grant No. 2024YFF0505703), Beijing Municipal Natural Science Foundation (Grant No. Z200021) and the National Natural Science Foundation of China (Grant No. 62088102).

## Impact Statement

This paper presents work whose goal is to advance the field of Machine Learning. There are many potential societal consequences of our work, none which we feel must be specifically highlighted here.

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

# A. Implementation Details

## A.1. Image Super-resolution Task

### A.1.1. DATA PREPARATION

We downloaded DIV2K dataset (Agustsson & Timofte, 2017) from this link, Set5 (Bevilacqua et al., 2012), Set14 (Zeyde et al., 2012), BSD100 (Martin et al., 2001b), Urban100(Huang et al., 2015), Manga109 (Matsui et al., 2017) from this link. Where DIV2K, Set5 and Set14 datasets already contain images with $\times 2$ downsampled using the cubic method. We used the cv2.INTER_CUBIC to generate $\times 2$ downsampled images for BSD100, Urban100, and Manga109 datasets. We convert each downsampled image to INR using SIREN (Sitzmann et al., 2020) with Adamax (Kingma & Ba, 2014) as optimizer with a learning rate of 1e-3 and 20,000 iterations. Specifically, to keep the INR representation accuracy of each INR consistent, we set the total number of parameters in the INR based on a percentage of the number of parameters in each image, and the percentage was set to 50%.

### A.1.2. TRAINING

For INSP (Xu et al., 2022), we implemented it based on their open-source code. And we expand the number of layers to 10, and the number of neurons per layer to 1024, making its MAC comparable to that of other methods. We use autograd to compute all $1^{st}$ to $3^{rd}$ order derivatives of INR at each points as described in (Xu et al., 2022). Then we set the input to be all $1^{st}$ to $3^{rd}$ order derivatives of INR at a point, and the output to be the rgb of a high-resolution image at that point. We trained 100 epochs with Adam (Kingma & Ba, 2014) optimizer at 0.001 learning rate after random initialization. For EDSR (Lim et al., 2017) and SwinIR (Liang et al., 2021), we implemented them based on link and link, largely maintaining the original hyperparameters. We trained them after random initialization. For our approach DVI, we set $P$ in the INR High Order Derivatives Computation module to 3. When using EDSR as the pre-existing network, we set the $K$ to 2 and select the outputs of conv_first layer and body layer in EDSR as intermediate features for fusion. When using SwinIR as the pre-existing network, we set the $K$ to 2 and select the outputs of conv_first layer and conv_after_body layer in SwinIR as intermediate features for fusion. We trained DVI with the pre-existing configuration (optimizer, learning rate, etc.). We use torchinfo to count the Trainable Parameters of all models and compute their MACs on an $156 \times 240$ image.

## A.2. Image Denosing Task

### A.2.1. DATA PREPARATION

We downloaded BSD500 dataset (Martin et al., 2001a) from this link, WED (Ma et al., 2016), from this link, CBSD68 (Martin et al., 2001a), Kodak24 (Franzen, 1999), and McMaster (Zhang et al., 2011) from this link. We used the cv2.add to add Gaussian noise with sigma of 15. We used only the first 1000 data in the WED dataset sorted by name. We convert each noisy image to INR using SIREN (Sitzmann et al., 2020) with Adamax (Kingma & Ba, 2014) as optimizer with a learning rate of 1e-3 and 20,000 iterations. Specifically, to keep the INR representation accuracy of each INR consistent, we set the total number of parameters in the INR same as the number of parameters in each image.

### A.2.2. TRAINING

For INSP (Xu et al., 2022), we implemented it based on their open-source code. And we expand the number of layers to 10, and the number of neurons per layer to 640, making its MAC comparable to that of other methods. We use autograd to compute all $1^{st}$ to $3^{rd}$ order derivatives of INR at each points as described in (Xu et al., 2022). Then we set the input to be all $1^{st}$ to $3^{rd}$ order derivatives of INR at a point, and the output to be the rgb of a clear image at that point. We trained 100 epochs with Adam(Kingma & Ba, 2014) optimizer at 0.001 learning rate after random initialization. For DnCNN (Zhang et al., 2017) and SwinIR (Liang et al., 2021), we implemented them based on link, largely maintaining the original hyperparameters. We trained them after random initialization. For our approach DVI, we set $P$ in the INR High Order Derivatives Computation module to 3. When using DnCNN as the pre-existing network, we set the $K$ to 2 and select the outputs of m_head layer and m_body layer in DnCNN as intermediate features for fusion. When using SwinIR as the pre-existing network, we set the $K$ to 2 and select the outputs of conv_first layer and conv_after_body layer in SwinIR as intermediate features for fusion. We trained DVI with the pre-existing configuration (optimizer, learning rate, etc.). We use torchinfo to count the Trainable Parameters of all models and compute their MACs on a $500 \times 500$ image.

### A.3. 3D Volume Segmentation Task

#### A.3.1. DATA PREPARATION

We downloaded Synapse dataset (Landman et al., 2015) from this link. We used the scipy.ndimage.zoom to scale down each volume with the corresponding label to $0.5\times$. We used the first 18 of the volumes for training and the last 12 for testing. We convert each volume to INR using SIREN (Sitzmann et al., 2020) with Adamax (Kingma & Ba, 2014) as optimizer with a learning rate of 1e-3 and 20,000 iterations. Specifically, to keep the INR representation accuracy of each INR consistent, we set the total number of parameters in the INR based on a percentage of the number of parameters in each volume, and the percentage was set to 20%.

#### A.3.2. TRAINING

For INSP (Xu et al., 2022), we implemented it based on their open-source code. And we expand the number of layers to 5, and the number of neurons per layer to 180, making its MAC comparable to that of other methods. We use autograd to compute all $1^{st}$ to $3^{rd}$ order derivatives of INR at each points as described in (Xu et al., 2022). Then we set the input to be all $1^{st}$ to $3^{rd}$ order derivatives of INR at a point, and the output to be the segmentation label at that point. We trained 100 epochs with Adam(Kingma & Ba, 2014) optimizer at 0.001 learning rate after random initialization. For VNET (Milletari et al., 2016) and UNet3D (Çiçek et al., 2016), we implemented them based on link, largely maintaining the original hyperparameters. We trained them after random initialization. For our approach DVI, we set $P$ in the INR High Order Derivatives Computation module to 3. When using VNET as the pre-existing network, we set the $K$ to 2 and select the outputs of in_tr layer and up_tr32 layer in VNET as intermediate features for fusion. When using UNet3D as the pre-existing network, we set the $K$ to 2 and select the outputs of conv3d_c1_1 layer and norm_lrelu_upscale_conv_norm_lrelu_l3 layer in UNet3D as intermediate features for fusion. We trained DVI with the pre-existing configuration (optimizer, learning rate, etc.). We use torchinfo to count the Trainable Parameters of all models and compute their MACs on a $64 \times 64 \times 64$ volume.

### A.4. Video Deblurring Task

#### A.4.1. DATA PREPARATION

We downloaded GoPro dataset (Nah et al., 2017) from this link. We used cv2.INTER_LINEAR to resize each frame to $690 \times 360$. We used only the first 40 frames of each video. We convert each video to INR using SIREN (Sitzmann et al., 2020) with Adamax (Kingma & Ba, 2014) as optimizer with a learning rate of 1e-3 and 20,000 iterations. We allocated 12,000 KB parameters for each INR.

#### A.4.2. TRAINING

For INSP (Xu et al., 2022), we implemented it based on their open-source code. And we expand the number of layers to 6, and the number of neurons per layer to 512, making its MAC comparable to that of other methods. We use autograd to compute all $1^{st}$ to $3^{rd}$ order derivatives of INR at each points as described in (Xu et al., 2022). Then we set the input to be all $1^{st}$ to $3^{rd}$ order derivatives of INR at the same point in 5 consecutive frames, and the output to be the rgb at that point in the center frame. We trained 100 epochs with Adam (Kingma & Ba, 2014) optimizer at 0.001 learning rate after random initialization. For VDTR (Cao et al., 2023) and PVDNet (Son et al., 2021), we implemented them based on link and link, largely maintaining the original hyperparameters. Except for VDTR, we adjusted the patch_size to 128 to ensure its compatibility with our graphics card. We trained them after random initialization. For our approach DVI, we set $P$ in the INR High Order Derivatives Computation module to 3. When using VDTR as the pre-existing network, we set the $K$ to 2 and select the outputs of img2feats layer and feature_encoder layer in VDTR as intermediate features for fusion. When using PVDNet as the pre-existing network, we set the $K$ to 3 and select the outputs of d0 layer, d1 layer, and temp layer in PVDNet as intermediate features for fusion. We trained DVI with the pre-existing configuration (optimizer, learning rate, etc.). We use torchinfo to count the Trainable Parameters of all models and compute their MACs on the GoPro dataset.

### A.5. Video Optical Flow Estimation Task

#### A.5.1. DATA PREPARATION

We downloaded Sintel dataset (Butler et al., 2012) from this link. We used cv2.INTER_LINEAR to resize each frame to $512 \times 218$. We divided the Sintel Training data into the training and testing sets required for this experiment in a ratio of

14:9. We convert each video to INR using SIREN (Sitzmann et al., 2020) with Adamax (Kingma & Ba, 2014) as optimizer with a learning rate of 1e-3 and 20,000 iterations. We allocated 160 KB parameters for each INR.

### A.5.2. TRAINING

For INSP (Xu et al., 2022), we implemented it based on their open-source code. And we expand the number of layers to 6, and the number of neurons per layer to 512, making its MAC comparable to that of other methods. We use autograd to compute all $1^{st}$ to $3^{rd}$ order derivatives of INR at each points as described in (Xu et al., 2022). Then we set the input to be all $1^{st}$ to $3^{rd}$ order derivatives of INR at the same point in 3 consecutive frames, and the output to be the optical flow at that point. We trained 100 epochs with Adam (Kingma & Ba, 2014) optimizer at 0.001 learning rate after random initialization. For FlowFormer (Huang et al., 2022) and SepFlow (Zhang et al., 2021a), we implemented them based on link and link, largely maintaining the original hyperparameters. We adjusted image_size to [216, 480] for FlowFormer and image_size to [192, 448] for SepFlow to ensure the compatibility with our graphics card. We trained them after random initialization. For our approach DVI, we set $P$ in the INR High Order Derivatives Computation module to 3. When using FlowFormer as the pre-existing network, we used two sets of feature extraction fusion networks, one for the Cost Volume Encoder and the other for the Cost Memory Decoder. We set $K$ to 1 for both. The former uses the output of the channel_convertor layer in MemoryEncoder as an intermediate feature, and the latter uses the output of the context_encoder in FlowFormer as an intermediate feature. When using SepFlow as the pre-existing network, we used two sets of feature extraction fusion networks, one for fnet layer and the other for cnet layer. We set $K$ to 1 for both. We trained DVI with the pre-existing configuration (optimizer, learning rate, etc.). We use torchinfo to count the Trainable Parameters of all models and compute their MACs on the Sintel dataset.

## A.6. DVI is Robust to Various Pre-existing Network Architectures

### A.6.1. DATA ANALYSIS

We calculated the statistical significance of performance differences between our method and the respective pre-existing network by Two-Sample t-Test. For the image super-resolution task, we used the PSNR metric on BSD100 (Martin et al., 2001b). For the image denoising task, we used the PSNR metric on CBSD68 (Martin et al., 2001a). For the 3D volume segmentation task, we used the DSC metric on Synapse 'mean' (Landman et al., 2015), and trim=0.2 for VNET. For the video deblurring task, we used the PSNR metric on GoPro (Nah et al., 2017). For the video optical flow estimation task, we used the EPE metric on Sintel 'final_ambush_2' (Butler et al., 2012).

## A.7. Derivative Map Contains Task-Relevant Structural Features

### A.7.1. TRAINING

In the 'zero' setting, we use torch.zeros_like to replace the derivative map. In the 'random' setting, we use torch.rand_like to replace the derivative map. We retrained DVI in the 'zero' and 'random' settings. In the 'mismatched' setting, We used the trained DVI from the original setting.

## A.8. Derivative Computation Techniques

### A.8.1. DATA ANALYSIS

In the experiments on 3D volume segmentation task, we used DVI(VNET) and '0029' volume from Synapse (Landman et al., 2015) to calculate the total time (network inference time + derivative map calculation time). In the experiments on image super-resolution task, we used DVI(SwinIR) and 'barbara' image from Set14 (Zeyde et al., 2012) to calculate the total time (network inference time + derivative map calculation time). These two experiments were conducted on one GPU RTX3090.

# B. More Results

*Figure S1.* Visual comparisons for image super-resolution task on images 'img_093' and 'img_089' from Urban100 (Huang et al., 2015).

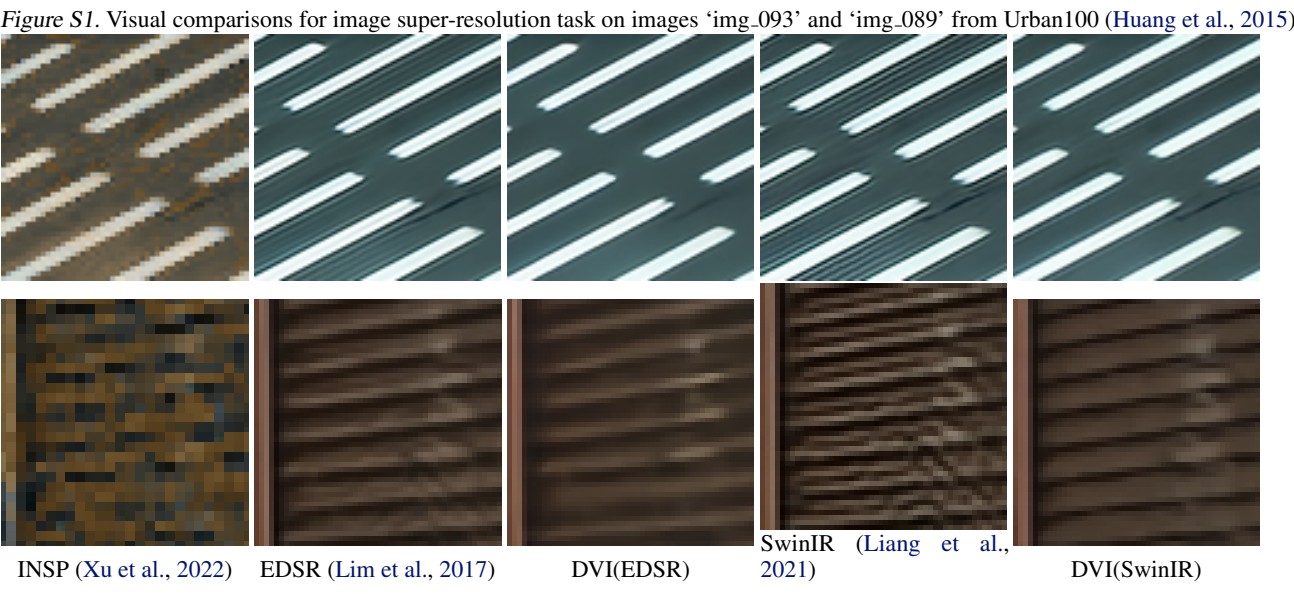

INSP (Xu et al., 2022)  EDSR (Lim et al., 2017)  DVI(EDSR)  SwinIR (Liang et al., 2021)  DVI(SwinIR)

*Figure S2.* Visual comparisons for image denoising task on images 'kodim01' and 'kodim17' from Kodak24 (Zhang et al., 2011).

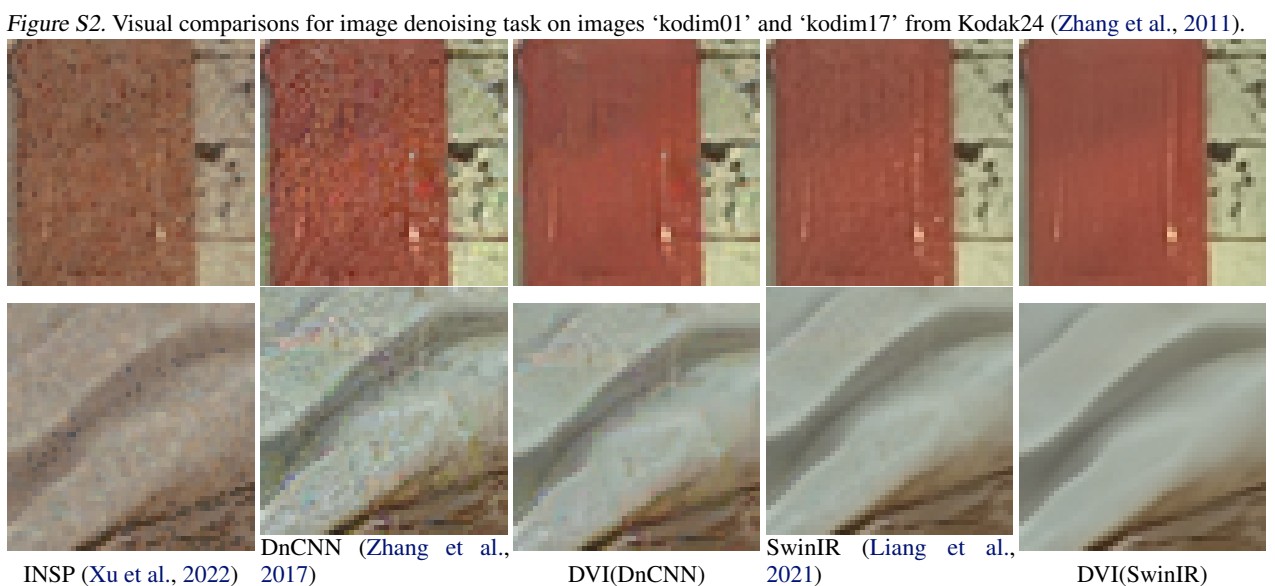

INSP (Xu et al., 2022)  DnCNN (Zhang et al., 2017)  DVI(DnCNN)  SwinIR (Liang et al., 2021)  DVI(SwinIR)

*Figure S3.* Visual comparisons for 3D volume segmentation task on data '0040' and '0034' from Synapse (Landman et al., 2015).

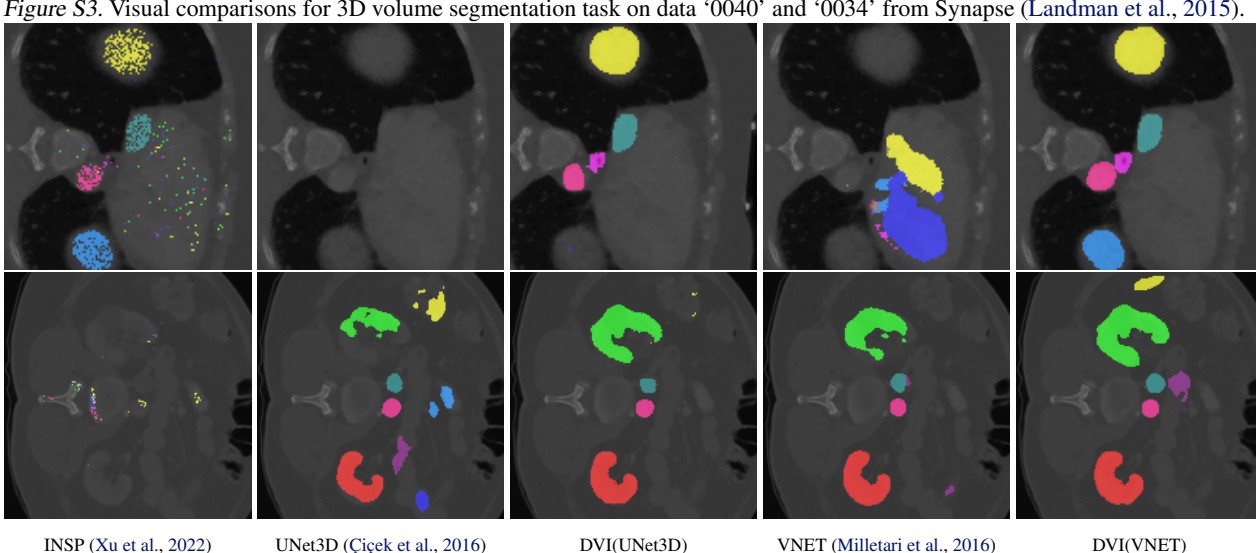

INSP (Xu et al., 2022)     UNet3D (Çiçek et al., 2016)     DVI(UNet3D)     VNET (Milletari et al., 2016)     DVI(VNET)

*Figure S4.* Visual comparisons for video deblurring task on videos 'GOPR0384_11_00' and 'GOPR0410_11_00' from GoPro (Nah et al., 2017).

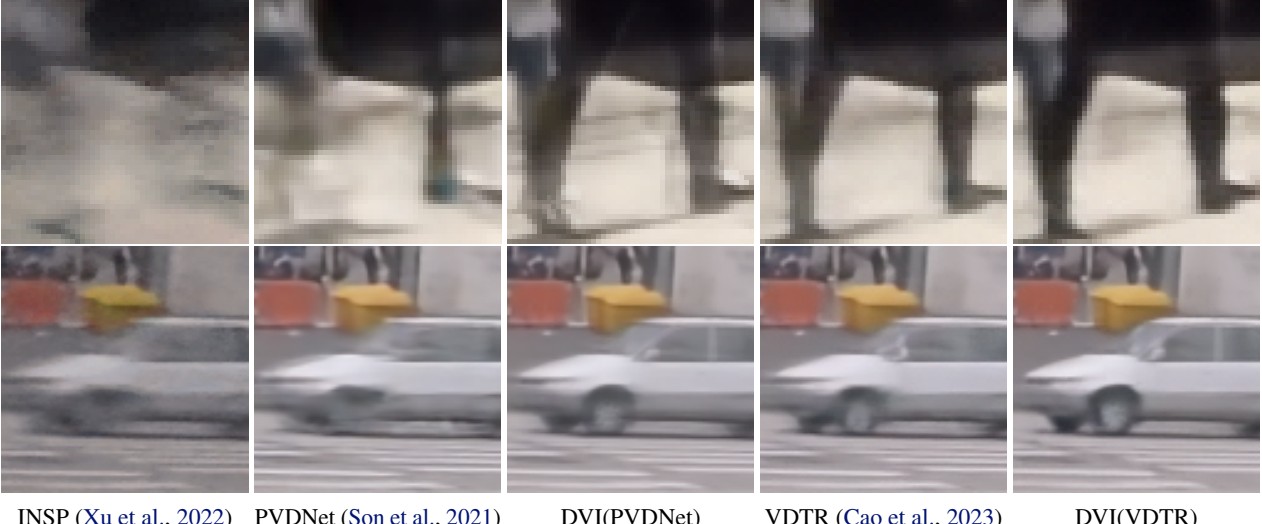

INSP (Xu et al., 2022)     PVDNet (Son et al., 2021)     DVI(PVDNet)     VDTR (Cao et al., 2023)     DVI(VDTR)

*Figure S5.* Visual comparisons for video optical flow estimation task on videos 'bandage' and 'market' from Sintel (Butler et al., 2012).

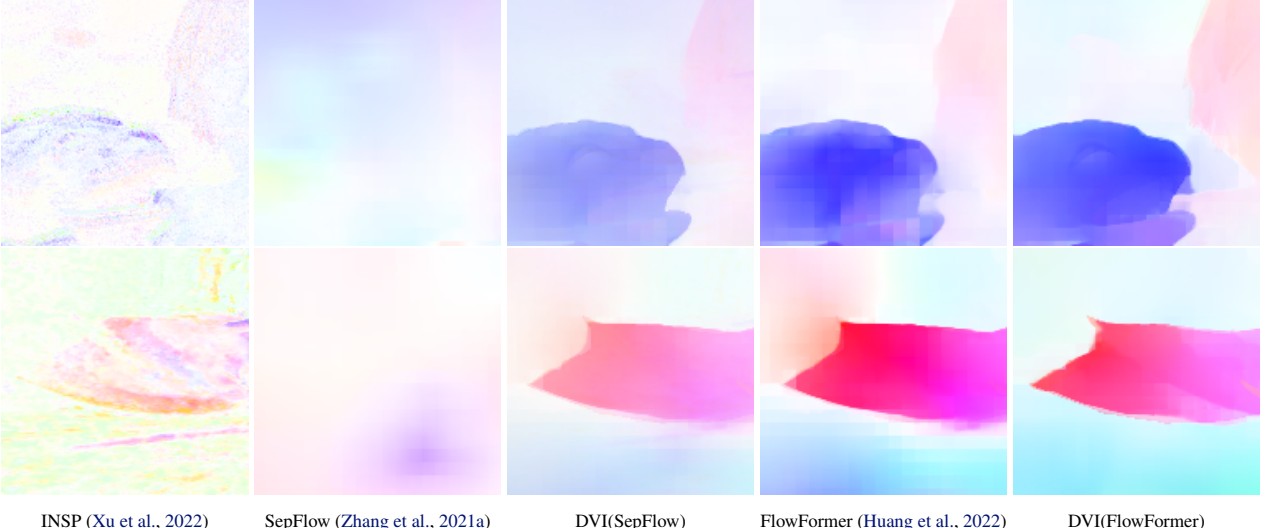

INSP (Xu et al., 2022)    SepFlow (Zhang et al., 2021a)    DVI(SepFlow)    FlowFormer (Huang et al., 2022)    DVI(FlowFormer)

*Figure S6.* Assessing the impact of substituting DVI's derivative map with alternative maps - zero-value (zero), random-value (random), and mismatched derivative (mismatch) - on the super-resolution task for the Urban100 dataset using SwinIR as the pre-existing network. On the left are bar plots for each alternative, with significant differences indicated (**: $p < 0.01$, ****: $p < 0.0001$). On the right are box plots showing the performance improvement of each alternative over the pre-existing network.

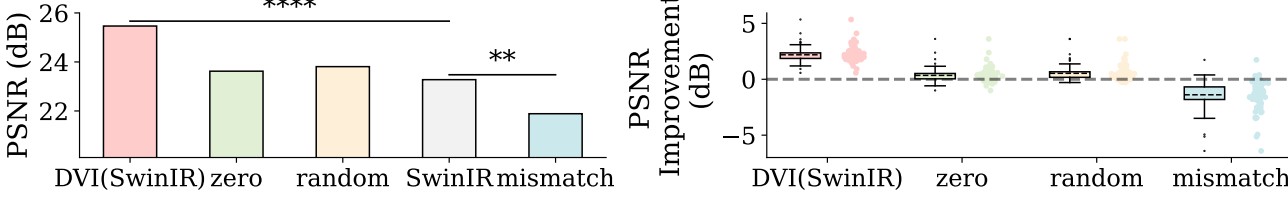

*Figure S7.* (a) The impact of removing the feature extraction and fusion modules from DVI on denoising task, CBSD68 dataset, with DnCNN as pre-existing network. (b) The comparison between DVI and the super-resolution sampling technique using INR (INR-SR) on super-resolution task, BSD100 dataset, with SwinIR as pre-existing network. Both subfigures have the same layouts as Figure 4.

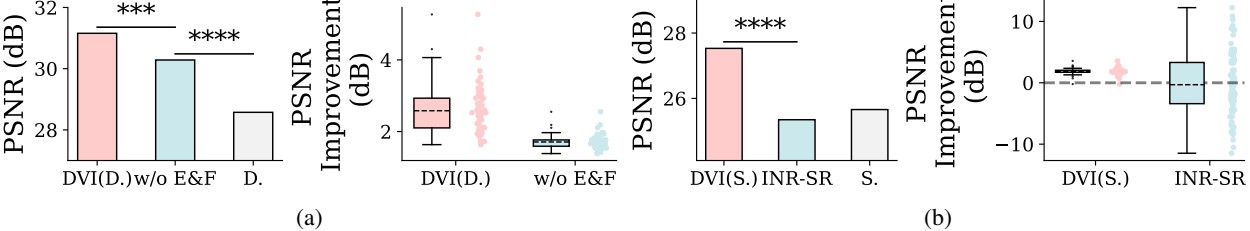

(a)                                    (b)

*Figure S8.* The performance comparison of DVI on image super-resolution task employing two distinct derivative computation techniques with SwinIR as pre-existing network. Left: barplots of each techniques. Right: curves of time (network inference time + derivative map calculation time) versus the highest order of the derivative map for each techniques in log scale.

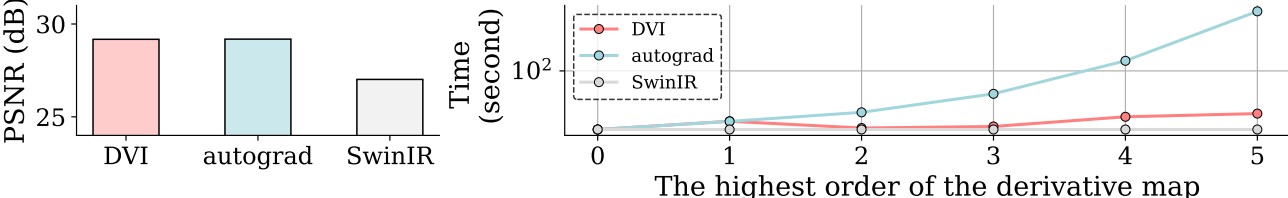

*Figure S9.* The curves of performance on video deblurring task (SSIM) (a) and video flow estimation task (Sintel clean dataset) (b) versus the highest order of the derivative map employed in DVI.

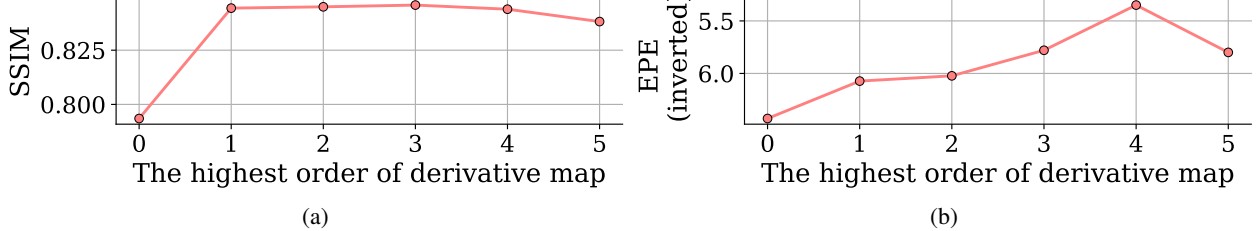

(a)    (b)

