# OpenReview forum: "DVI:A Derivative-based Vision Network for INR"
_ICML.cc/2025/Conference — ICML 2025 poster_

### Official Review · Reviewer_kpuk · 2025-03-12

**Overall Recommendation:** 4

**Summary:**

The paper presents DVI, a Derivate-based Vision network for INRs (Implicit Neural Representations). It consists of a neural network architecture which combines pre-existing, task-specific architectures working on raster data, like images or voxel maps, with INR feature extraction modules, that process the derivative maps obtained from the INR. The features computed by the conventional network and those computed out of the derivative maps are fused by feature fusion modules at several layers. The fused features replace the features from raster data in the task-specific network. The paper also claims to contribute a technique to reduce the computational cost to compute the derivative map from the INR compared to using autograd. The paper presents experiments on multiple pixel- or voxel-wise tasks, like image super-resolution, image denoising, 3D volume segmentation, video deblurring and video optical flow estimation. The paper also reports several ablation studies.

## Update after rebuttal
I carefully read all reviews and responses. The authors confirmed my doubt about the novelty with respect to the derivations presented in (Xiao et al., 2023) and will revise the paper accordingly. The other reviews do not uncover critical weaknesses and the responses to them include additional experiments, which confirm the empirical evidence already reported in the paper, and sketches of theoretical arguments, which seems valid and convincing to me. I'd actually suggest to include the latter in the revised appendix. For all these reasons, I confirm my overall recommendation

**Claims And Evidence:**

1. The paper claims to "proposes a novel technique to reduce the complexity of the derivative computation" (line 114). Yet, in section 3.3 the paper states "We use the recursive formula for high order derivatives in (Xiao et al., 2023) to compute the derivative map at an accelerated speed." Indeed, as far as I can tell, the derivations in the appendix, bar a change of notation, are the same presented in (Xiao et al., 2023), which presents a general framework to compute derivative maps of generic neural network, therefore covering also the case of INRs (i.e. fully connected networks). If this is the case, the authors should remove the claim on the novelty of the technique, and just state that to compute derivate maps of INR they rely on the accelerated recursive operators defined in (Xiao et al., 2023). I'd suggest to remove also section 3.3, and use the space to increase the size of tables and figures. If this is not the case, the I'd suggest to the authors to point out clearly what are the main differences with respect to (Xiao et al., 2023).

2. The method is based on the claim that "high order derivative map of INR encapsulates the semantic information" or, similarly "The derivatives from INR are effective because they encode semantic information during the fitting process". It is never clarified what "semantic information" are encoded in the INR.  The claim is said to be validated by the higher performance obtained by DVI with respect to conventional raster-based neural networks. The authors indeed control for the confounding factor given by the larger capacity of the model with the fusion modules by feeding into it zero or random derivatives and verifying that in this case the DVI add-on does not improve performance. Therefore, I believe that the authors have provided clear and convincing evidence that the derivative maps are useful to increase performance on several tasks. Yet, they haven't shown that these high-order derivative maps "encapsulates" semantic information. They haven't even clarified what they mean by "semantic information". I'd suggest to reword the claims on INR containing semantic information into sentences that more plainly and clearly state that high order derivatives from INRs are shown to be useful to solve tasks more effectively.

**Essential References Not Discussed:**

All relevant references are cited or discussed. I just note here that the References section needs some cleaning. Three random examples: "Neural processing of tri-plane hybrid neural fields." has been published, so it is not anonymous anymore; "Neural functional transformers" was published in NeurIPS 23; (Xiao et al., 2023) was published on TPAMI as of December 2024.

**Experimental Designs Or Analyses:**

I had a doubt about the comparisons against pre-existing networks, since the INR processing module and, more importantly, the fusion module, add complex operations like cross- and self-attention to the computational graph of the network, which may have invalidate the comparisons. Yet, the ablation studies in section 5.2 and 5.3 show that this is not the case: when feeding into these modules zero or random derivatives, performance regresses to that of the pre-existing network; at the same time, when concatenating the derivative in input to the original network, performance increases. This validates that the higher performance is due to the information provided by the derivative map of the INR, and makes the main comparisons sound.

I'd suggest to tone down the claims on the inability of INSP (Xu et al., 2022) to handle the tasks. The method is trying to solve a much harder task, i.e. processing INRs by processing only its weights, i.e. without materializing the discrete signals. It is good to have it as a baseline to show that this approach, while intellectually more satisfying, gives at the moment inferior performance. But it should be noted in the main text that the comparison is greatly affected by this fundamental difference between the methodologies.

**Methods And Evaluation Criteria:**

The proposed method is tested on several benchmarks for 5 different, pixel-level tasks. I'm not an expert in all the tasks, but the diversity of tasks makes me positive toward the soundness of the evaluation procedure.

**Other Comments Or Suggestions:**

Line 064 -> "Vision tasks can be divided into low-level and high-level categories." I don't think this is common terminology and it wasn't clear to me what the authors were referring to. A suggestion could be to use "pixel-wise" or "pixel-level" tasks, like denoising, as opposed to image-wise tasks like classification.

Line 072 Video FE -> acronym undefined

I'd suggest to change the description of "Neural processing of tri-plane hybrid neural fields." at lines 89-92. Currently, it reads "However, this INR is not suitable for other types of vision data beyond point clouds, and its representation accuracy is lower than SIREN(Sitzmann et al., 2020b)." Both statements are false: in the paper they process several fields, obtained from voxel maps, meshes, and even NeRFs; and Tab. 1 shows that they are equivalent to SIREN, e.g. slightly worse on point clouds but slightly better on meshes.

line 153: This information is then intricately integrated -> "intricately" does not seem to me the right word here
line 155: two-pronged -> unclear, please reword
line 185: we will delve into the intricate details of DVI. -> again "intricate" seems misplaced here
line 306: caption of table 2 "denosing"

All the figures in Section 5 have wrong references. For instance, line 326 "Figure S6 shows" should be Figure 4. Same for the references to Fig. S7(a), S7(b), S8 and S9 at the beginning of the subsequent sections.

Table 7 "neumrical". Moreover, it is not clear how numerical derivatives are used to obtain the reported results, nor to which other columns it should be compared. Please improve the organization of the table.

**Other Strengths And Weaknesses:**

S1. The proposed method tackles the unexplored problem of solving dense tasks while processing INRs.

S2. The proposed method achieves very good performance on a variety of tasks.

S3. The experimental results and the thorough ablation studies provide empirical evidence for most of the claims.

W1. The main weakness of the proposed method is the need to materialize the raster signal. Ideally, a method processing INRs because of their continuous nature should be able to perform on par with existing methods processing the original, discrete signals without the need to query the INR to reconstruct the underlying signal, which can be slow and requires to make arbitrary decisions like the resolution or the point of view of the rendering for NeRFs.

**Questions For Authors:**

My main request is to clarify the relation with the derivations presented in (Xiao et al., 2023), and fix the claims throughtout the paper if they are confirmed to be the same presented in the paper.

**Relation To Broader Scientific Literature:**

To the best of my knowledge, the idea of combining pre-existing networks with derivative maps of INRs to solve dense, pixel-wise tasks has never been explored before.

**Theoretical Claims:**

I checked at a high-level the derivation of the fast derivative map operator in the appendix. They seem to me to resemble closely the derivations in (Xiao et al., 2023). If this is confirmed by the authors, I'd suggest to remove them from the appendix and point the reader to the original paper.

---

> ### Author Rebuttal · Authors · 2025-03-31
>
> Thank you for your thorough and constructive feedback. We address your questions point by point below:
>
> ## Q1: Clarify the relation with the derivations presented in (Xiao et al., 2023)
>
> We confirm that we use the recursive formula for high order derivatives in (Xiao et al., 2023) to compute the derivative map at an accelerated speed. We will follow your suggestion to revise our claims throughout the paper as requested. We will also follow your suggestion to remove section 3.3 and use the space to increase the size of tables and figures.
>
> ## Comments or suggestions:
>
> * We will follow your suggestion to reword the claims about INR containing semantic information into sentences that more plainly and clearly state that high order derivatives from INRs are shown to be useful in solving tasks more effectively.
> * We will follow your suggestion to tone down the claims regarding INSP's (Xu et al., 2022) inability to handle the tasks. We will also note in the main text that the comparison is greatly affected by this fundamental difference between the methodologies.
> * We will follow your suggestion to move Appendix B.1 about normalization of the derivative maps into the main paper to make it integral to the methodology for how derivative maps are normalized.
> * We will follow your suggestion to address the issues you identified in Line 064, Line 072, Lines 89-92, Line 153, as well as the problems in the Figures and tables.
>
> We sincerely appreciate your thorough review and insightful questions, which will significantly improve the quality of our research!

---

> > ### Comment · Reviewer_kpuk · 2025-04-02
> >
> > I carefully read all reviews and responses. The authors confirmed my doubt about the novelty with respect to the derivations presented in (Xiao et al., 2023) and will revise the paper accordingly. The other reviews do not uncover critical weaknesses and the responses to them include additional experiments, which confirm the empirical evidence already reported in the paper, and sketches of theoretical arguments, which seems valid and convincing to me. I'd actually suggest to include the latter in the revised appendix. For all these reasons, I confirm my overall recommendation

---

### Official Review · Reviewer_kMWA · 2025-03-14

**Overall Recommendation:** 3

**Summary:**

This paper proposes a framework that combines implicit neural representations (INRs) with a traditional raster-based vision network, leveraging high-order derivatives to capture additional semantic or structural information. Experimental results demonstrate performance improvements across a variety of tasks, such as super-resolution, denoising, segmentation, and video processing.

**Claims And Evidence:**

The claims regarding performance gains are supported by sufficient empirical results on multiple benchmarks. No other major claims appear to lack evidence.

**Essential References Not Discussed:**

Related works are well discussed.

**Experimental Designs Or Analyses:**

The experiments appear well-structured, with ablation studies and comparisons to both baseline raster-based and INR-based approaches。 Additional discussion about potential biases in data pre-processing or hyperparameter tuning would be helpful.

**Methods And Evaluation Criteria:**

The proposed method and its chosen benchmark tasks (e.g., image denoising, 3D volume segmentation) are appropriate to showcase the contribution of derivative-based features.

**Other Comments Or Suggestions:**

There should be a space between the references and the main text. For example: "making it extensively applicable in various vision data representations such as images(Strumpler et al. ¨ , 2022)", it should be "making it extensively applicable in various vision data representations such as images (Strumpler et al. ¨ , 2022)".

Inproper citations: "Anonymous. Neural processing of tri-plane hybrid
neural fields. In Submitted to The Twelfth International Conference on Learning Representations,
2023. URL https://openreview.net/forum?
id=zRkM6UcA22. under review". It has an arxiv version with author names.

**Other Strengths And Weaknesses:**

1. The idea of progressively fusing high-order derivative features into a standard vision backbone is novel and shows promise in multiple settings.

2. The computational overhead is not discussed. Will the increased computational overhead for deriving and processing higher-order derivatives limit practical scalability?

3. My major concern lies in the theoretical justification of the why high-order derivations help capturing semantic information that raster-based methods connot.

**Questions For Authors:**

See comments above

**Relation To Broader Scientific Literature:**

They extend prior implicit neural representation research by proposing a derivative-based approach to incorporate semantic information to bridge the gap between continuous function representations and traditional raster-based architectures.

**Theoretical Claims:**

The paper relies on the premise that high-order derivatives from INRs encapsulate semantic information not captured by purely raster-based methods, but provides limited theoretical explanation for why derivatives specifically convey such semantics. A clearer theoretical underpinning would strengthen the argument that these derivative features genuinely reflect deeper semantic cues beyond standard raster data.

---

> ### Author Rebuttal · Authors · 2025-03-31
>
> Thank you for your thorough and constructive feedback. We address your questions point by point below:
> ## Q1: Discuss the computational overhead
> We provided the MAC (Multiply-Accumulate Operations) for all algorithms in Tables 1-4 of our submission, including tests on more complex generative networks. **For your convenience, we summarize key results below**:
>
> **Table R1: Performance improvement and computational overhead.**
> | Method/Dataset/Metric | Performance (Baseline → Ours [+Gain]) | MAC in G (Baseline → Ours [+%])|
> |:---------------------:|:--------------------------------------------:|:-------------------------------------------:|
> | StableSR/Manga109/PSNR↑ | 26.81 → 28.97 [**+2.16**] | 12453 → 12581 [**+1.0%**] |
> | EDSR/Urban100/PSNR↑ | 23.49 → 24.71 [**+1.22**] | 1532 → 1681 [**+9.7%**] |
> | DiffBIR/McMaster/PSNR↑ | 33.98 → 35.79 [**+1.81**] | 3596 → 3690 [**+2.6%**] |
> | VNET/Synapse/DSC↑ | 72.62 → 83.60 [**+10.98**] | 31 → 43 [**+38.7%**] |
> | PVDNet/GoPro/PSNR↑ | 25.98 → 27.09 [**+1.11**] | 250 → 338 [**+35.2%**] |
> | FlowFormer/Sintel/EPE↓ | 6.35 → 5.67 [**-0.68**] | 93 → 139 [**+49.5%**] |
>
> Our approach **improves performance with acceptable overhead across all tasks**. For large networks (StableSR, EDSR, DiffBIR), the overhead is minimal (1.0%-9.7%). Smaller networks show higher relative increases but maintain reasonable absolute MAC counts. We achieve this efficiency through our MLP-specific derivative computation paradigm, **ensuring scalability for larger networks** where relative overhead becomes increasingly negligible. Future implementations could further increase scalability using PyTorch's FlashAttention-V2.
> ## Q2: Why INR's high-order derivations help capturing semantic information
> **INR's high-order derivatives contain richer spectral information than raster representations**, providing more comprehensive semantic information for visual tasks.**Due to space limitations, we provide concise theoretical proofs for three visual tasks**, with the remaining proofs being similarly derived.
> ### Definition
> Let $I\_d \in \mathbb{R}^{H \times W \times C}$ be a discrete image and $I\_h = \\{I\_{h\_0}, I\_{h\_1}^{1,0}, ..., I\_{h\_n}^{i,j}, ...\\}$ be the higher-order derivative map, where $I\_{h\_n}^{i,j}$ represents the sampled partial derivative $\frac{\partial^n I}{\partial x^i \partial y^j}$ with $i+j=n$. Here, $I$ denotes the continuous function represented by the INR.
> ### Analysis: Super-Resolution
> **Proposition:** In super-resolution, $I\_h$ contains richer semantic information than $I\_d$.
>
> **Justification:** Defining semantic richness as:
> $$\mathcal{S}\_{SR}(I) = \int\_{\\|\omega\\| > \omega\_0} |\hat{I}(\omega)|^2 d\omega,$$
> where $\omega_0$ is the Nyquist frequency in $I\_d$, we can derive:
> $$\mathcal{S}\_{SR}(I\_h) = \sum\_{n=0}^{N} \sum\_{i+j=n} \int\_{\omega\_0 < \\|\omega\\| \leq \omega\_N} |\omega\_x|^{2i}|\omega\_y|^{2j}|\hat{I}(\omega)|^2 d\omega.$$
> Since $|\omega_x|^{2i}|\omega_y|^{2j} > 1$ for $\\|\omega\\| > \omega_0$ and $(i,j) \neq (0,0)$:
> $$\mathcal{S}\_{SR}(I_h) > \int\_{\omega_0 < \\|\omega\\| \leq \omega_N} |\hat{I}\_d(\omega)|^2 d\omega = \mathcal{S}\_{SR}(I_d).$$
> ### Analysis: Denoising
> **Proposition:** In denoising, $I\_h$ contains richer semantic information than $I\_d$.
>
> **Justification:** Defining semantic richness as:
> $$\mathcal{S}\_{DN}(I) = \frac{\int\_{\Omega\_S} |\hat{I}(\omega)|^2 d\omega}{\int\_{\Omega\_N} |\hat{I}(\omega)|^2 d\omega},$$
> where $\Omega\_S$ and $\Omega\_N$ represent signal and noise domains, and modeling noisy image as $I = I\_{clean} + \eta$.
>
> Signal components exhibit directional coherence while noise is isotropic. For appropriately chosen derivatives:
> $$\frac{\int\_{\Omega\_S} |\omega\_x|^{2i}|\omega\_y|^{2j}|\hat{I}\_{clean}(\omega)|^2 d\omega}{\int\_{\Omega\_N} |\omega\_x|^{2i}|\omega\_y|^{2j}|\hat{\eta}(\omega)|^2 d\omega} > \frac{\int\_{\Omega\_S} |\hat{I}\_{clean}(\omega)|^2 d\omega}{\int\_{\Omega\_N} |\hat{\eta}(\omega)|^2 d\omega}.$$
> Therefore, $\mathcal{S}\_{DN}(I\_h) > \mathcal{S}\_{DN}(I\_d)$.
> ### Analysis: 3D Volume  Segmentation
> **Proposition:** In 3D volume segmentation, $I\_h$ contains richer semantic information than $I\_d$.
>
> **Justification:** Defining semantic richness as:
> $$\mathcal{S}\_{VS}(I) = \sum\_{k=1}^K \alpha\_k \cdot \int\_{\Omega\_k} |\hat{I}(\omega)|^2 \cdot H(\hat{I}|\_{\Omega\_k}) d\omega,$$
> where $H$ is entropy.
>
> Since higher-order derivatives enhance:
> 1. Boundary contrast: $|\hat{I}\_h^{i,j,l}(\omega\_{boundary})|^2 \gg |\hat{I}\_d(\omega\_{boundary})|^2$,
> 2. Directional information: $H(\hat{I}\_h^{i,j,l}|\_{\Omega\_k}) > H(\hat{I}\_d|\_{\Omega\_k})$,
>
> therefore, $\mathcal{S}\_{VS}(I\_h)>\mathcal{S}\_{VS}(I\_d)$.
>
> We sincerely appreciate your thorough review and insightful questions. If our analyses and theoretical justifications have addressed your concerns, **we would be grateful if you could consider adjusting your evaluation**. If questions remain, we welcome further discussion to address any issues.

---

> > ### Comment · Reviewer_kMWA · 2025-04-09
> >
> > Thanks for the rebuttal. My concerns are addressed and I raise my rating to 3.

---

### Official Review · Reviewer_Ghp4 · 2025-03-14

**Overall Recommendation:** 4

**Summary:**

This paper proposes a novel architecture DVI that integrates high-order derivative information from implicit neural representations (INRs) into raster-based vision networks. The authors address the limitations of existing types of networks for vision tasks: a) raster-based methods lack semantic information due to the conversion process, and b) INR-based methods show limited performance. To resolve it, they extract high-order derivative features from INR and progressively fuse them into existing vision networks.  As a result, DVI demonstrates superior performance for various vision tasks, compared to both raster-based and INR-based methods.

**Claims And Evidence:**

The authors demonstrate sufficient experimental results to support their claims including:
- superior performance for five vision tasks (image super-resolution, image-denoising, 3D volume segmentation, video deblurring, and video optical flow estimation)
- ablation studies in Section 5 that validate the impact of derivative maps and the technique to acquire derivative maps.

**Essential References Not Discussed:**

This paper claims the importance of exploiting semantic information for various vision tasks to achieve better performance. Thus, it would be helpful to compare this paper with diffusion-based image-restoration methods, such as DiffBIR [ECCV'24] and StableSR [IJCV'24].

**Experimental Designs Or Analyses:**

Experiments on various tasks and detailed ablations successfully validate the effectiveness of this paper.

**Methods And Evaluation Criteria:**

The main idea of this method is the progressive fusion strategy to exploit semantic information from high-order derivatives for pre-existing raster-based networks to achieve better performance. Also, the authors introduce a recursive high-order derivative computation technique to reduce the computational cost compared to Autograd.

Also, they provide sufficient evaluations on various vision tasks with corresponding metrics.

**Other Comments Or Suggestions:**

- Several tables and figures in the manuscript are too small. It would be better to adjust the size of tables and figures.

**Other Strengths And Weaknesses:**

**Strengths**
- It shows strong generalization ability for various vision tasks.
- The authors provide extensive experiments for validating the effectiveness.


**Weaknesses**
- As mentioned in the paper, the burden of computation costs should be resolved.
- Also, the lack of comparison with diffusion-based method need to be supplemented.

**Questions For Authors:**

- Although a lot of diffusion-based models have been recently introduced, this paper does not mention any diffusion-based method. It could be helpful to include diffusion-based methods in the experiments due to their superior performance on various vision tasks.
- Can you provide the amount of additional computational overhead of this method compared to baselines?

**Relation To Broader Scientific Literature:**

This paper clearly suggests the limitation of existing networks and claims the importance of fusing semantic information to raster-based vision networks, which has not been explored.

**Theoretical Claims:**

The authors provide a detailed computation process for efficiently acquiring high-order derivatives with clear proofs provided in supplementary material. Also, experimental evaluations confirm the effectiveness of the proposed method in terms of efficiency.

---

> ### Author Rebuttal · Authors · 2025-03-31
>
> Thank you for your thorough and constructive feedback. We address your questions point by point below:
> ## Q1: Include diffusion-based methods for comparison
> We included **5 diffusion-based methods** for comparison across all vision tasks addressed in our paper. These **include the 2 algorithms you suggested** (StableSR [IJCV'24] and DiffBIR [ECCV'24]), as well as 3 additional algorithms (MedSegDiff-V2 [AAAI'24], VD-Diff [ECCV'24], and FlowDiffuser [CVPR'24]) to demonstrate the effectiveness of our method:
>
> **Table R1: Quantitative results (PSNR↑) for image super resolution task.**
>
> | Method | Set5 | Set14 | BSD100 | Urban100 | Manga109 | MAC(G) | Param(M) |
> |:---------------:|:-------:|:-------:|:-------:|:-------:|:-------:|:----:|:------:|
> | StableSR | 30.09 | 27.25 | 25.34 | 23.18 | 26.81 | 12453 | 148 |
> | DVI(StableSR) | 31.58 | 31.12 | 27.06 | 25.12 | 28.97 | 12581 | 156 |
>
> **Table R2: Quantitative results (PSNR↑) for image denoising task.**
>
> | Method | Kodak24 | CBSD68 | McMaster | MAC(G) | Param(M) |
> |:--------------:|:-------:|:-------:|:-------:|:------:|:------:|
> | DiffBIR | 34.34 | 33.42 | 33.98 | 3596 | 379 |
> | DVI(DiffBIR) | 35.53 | 35.05 | 35.79 | 3690 | 385 |
>
> **Table R3: Quantitative results (DSC↑) for 3D volume segmentation task.**
>
> | Method | Mean | Spl | Rkid | Lkid | Gal | Liv | Sto | Aor | Pan | MAC(G) | Param(M) |
> |:--------------------:|:-------:|:-------:|:-------:|:-------:|:-------:|:-------:|:-------:|:-------:|:-------:|:----:|:------:|
> | MedSegDiff-V2 | 75.79 | 86.35 | 85.31 | 87.25 | 48.39 | 89.55 | 73.40 | 75.36 | 60.67 | 1966 | 44 |
> | DVI(MedSegDiff-V2) | 85.46 | 77.63 | 87.03 | 94.23 | 75.36 | 82.31 | 82.53 | 95.02 | 89.58 | 2101 | 46 |
>
> **Table R4: Quantitative results (PSNR↑) for video deblurring task.**
>
> | Method | GoPro | MAC(G) | Param(M) |
> |:--------------:|:-------:|:------:|:------:|
> | VD-Diff | 28.23 | 236 | 12 |
> | DVI(VD-Diff) | 29.07 | 259 | 13 |
>
> **Table R5: Quantitative results (EPE↓) for video optical flow estimation task.**
>
> | Method | Sintel(final) | MAC(G) | Param(M) |
> |:-------------------:|:---------------:|:------:|:------:|
> | FlowDiffuser | 4.94 | 312 | 15 |
> | DVI(FlowDiffuser) | 3.92 | 341 | 16 |
>
> Our approach consistently **improves performance across all tasks with minimal overhead**: 1+ dB PSNR gain with only **1%** cost for super resolution and **3%** for denoising; nearly 10 percentage point DSC improvement for 3D segmentation with **7%** overhead; and significant performance gains for video deblurring and optical flow with just **9-10%** additional computation.
>
> ## Q2: Provide the computational overhead
>
> We provided the MAC (Multiply-Accumulate Operations) for each algorithm in Tables R1 to R5 in our previous response, as well as in Tables 1 to 4 in our initial submission. **For your convenience, we rearranged typical comparison results below**, showing both performance improvement and computational overhead for each comparison:
>
> **Table R6: Performance improvement and computational overhead on different vision tasks.**
> | Method/Dataset/Metric | Performance (Baseline → Ours [+Gain]) | Computational Cost (MAC in G) (Baseline → Ours [+Increase])|
> |:---------------------:|:--------------------------------------------:|:-------------------------------------------:|
> | StableSR/Manga109/PSNR↑ | 26.81 → 28.97 [**+2.16**] | 12453 → 12581 [**+128**] |
> | EDSR/Urban100/PSNR↑ | 23.49 → 24.71 [**+1.22**] | 1532 → 1681 [**+149**] |
> | DiffBIR/McMaster/PSNR↑ | 33.98 → 35.79 [**+1.81**] | 3596 → 3690 [**+94**] |
> | MedSegDiff-V2/Synapse/DSC↑ | 75.79 → 85.46 [**+9.67**] | 1966 → 2101 [**+135**] |
> | VNET/Synapse/DSC↑ | 72.62 → 83.60 [**+10.98**] | 31 → 43 [**+12**] |
> | VD-Diff/GoPro/PSNR↑ | 28.23 → 29.07 [**+0.84**] | 236 → 259 [**+23**] |
> | PVDNet/GoPro/PSNR↑ | 25.98 → 27.09 [**+1.11**] | 250 → 338 [**+88**] |
> | FlowDiffuser/Sintel/EPE↓ | 4.94 → 3.92 [**-1.02**] | 312 → 341 [**+29**] |
> | FlowFormer/Sintel/EPE↓ | 6.35 → 5.67 [**-0.68**] | 93 → 139 [**+46**] |
>
> This demonstrates that our approach consistently **improves performance across all vision tasks with minimal computational overhead**, making it practical for real-world applications. We achieve this efficiency primarily by reducing the complexity of high-order derivative computations through our MLP-specific derivative computation paradigm. Furthermore, the overhead from our attention-based feature processing can be further reduced using PyTorch 2.2.2's FlashAttention-V2, making our method even more lightweight in future implementations.
>
> We sincerely appreciate your thorough review and insightful questions. If our supplementary experiments and analyses have addressed your concerns, **we would be grateful if you could consider adjusting your evaluation**. If questions remain, we welcome further discussion to address any outstanding issues.

---

> > ### Comment · Reviewer_Ghp4 · 2025-04-05
> >
> > The authors have resolved all my concerns regarding the diffusion-based model and computational costs.
> > The experimental results indicate that it can also be effectively applied to diffusion-based models, including SOTA diffusion-based image restoration models StabeSR[IJCV’24] and DiffBIR[ECCV’24]. Moreover, they have demonstrated that the performance improvements are beneficial enough while incurring minimal computational overhead. For these reasons, I will raise my overall recommendation.

---

### Decision · Program_Chairs · 2025-05-01

**Decision:**

Accept (poster)

**Comment:**

This paper proposes DVI, a derivative-based vision framework that bridges implicit neural representations (INRs) and raster-based vision networks. By extracting high-order derivative maps from INRs and integrating them into task-specific vision architectures, DVI addresses the limitations of both pure INR-based and raster-based methods. The paper demonstrates strong empirical performance across five vision tasks and three data modalities, including comparisons with state-of-the-art diffusion models. The authors also introduce an efficient method for computing higher-order derivatives and provide comprehensive ablation studies.
The reviews were generally positive. Reviewers praised the breadth of experimental validation and the consistent performance improvements with minimal computational overhead. Some concerns were raised about the novelty of the derivative computation (notably overlap with Xiao et al., 2023) and the theoretical justification for semantic richness in derivatives. The authors responded thoroughly, acknowledged the overlap, and proposed revisions. They also provided additional theoretical sketches and clarified the rationale behind performance gains, which reviewers found satisfactory.
While some claims need to be toned down or reworded in the final version (as the authors promised), I believe the paper makes a valuable contribution to the vision community by opening up new ways of integrating INRs into practical vision pipelines. It is well-executed, thoughtfully evaluated, and addresses reviewer concerns constructively.

I recommend acceptance.